# Development of Novel ROCK Inhibitors via 3D-QSAR and Molecular Docking Studies: A Framework for Multi-Target Drug Design

**DOI:** 10.3390/pharmaceutics16101250

**Published:** 2024-09-26

**Authors:** Milan Beljkas, Milos Petkovic, Ana Vuletic, Ana Djuric, Juan Francisco Santibanez, Tatjana Srdic-Rajic, Katarina Nikolic, Slavica Oljacic

**Affiliations:** 1Department of Pharmaceutical Chemistry, Faculty of Pharmacy, University of Belgrade, Vojvode Stepe 450, 11221 Belgrade, Serbia; milan.beljkas@pharmacy.bg.ac.rs (M.B.); slavica.oljacic@pharmacy.bg.ac.rs (S.O.); 2Department of Organic Chemistry, Faculty of Pharmacy, University of Belgrade, Vojvode Stepe 450, 11221 Belgrade, Serbia; milos.petkovic@pharmacy.bg.ac.rs; 3Department of Experimental Oncology, Institute for Oncology and Radiology of Serbia, Pasterova 14, 11000 Belgrade, Serbia; radovanovica@ncrc.ac.rs (A.V.); ana.djuric@ncrc.ac.rs (A.D.); 4Group for Molecular Oncology, Institute for Medical Research, National Institute of the Republic of Serbia, University of Belgrade, Dr. Subotica 4, 11129 Belgrade, Serbia; jfsantibanez@imi.bg.ac.rs

**Keywords:** Rho-associated protein kinases, multitarget-directed ligands, rational drug design, molecular modeling, 3D-QSAR, antineoplastic agents, breast cancer, pancreatic ductal adenocarcinoma

## Abstract

**Background/Objectives:** Alterations in the actin cytoskeleton correlates to tumor progression and affect critical cellular processes such as adhesion, migration and invasion. Rho-associated coiled-coil-containing protein kinases (ROCK1 and ROCK2), important regulators of the actin cytoskeleton, are frequently overexpressed in various malignancies. The aim of this study was therefore to identify the key structural features of ROCK1/ROCK2 inhibitors using computer-aided drug design (CADD) approaches. In addition, new developed ROCK inhibitors provided a significant framework for the development of multitarget therapeutics—ROCK/HDAC (histone deacetylases) multitarget inhibitors. **Methods**: 3D-QSAR (Quantitative structure-activity relationship study) and molecular docking study were employed in order to identify key structural features that positively correlate with ROCK inhibition. MDA-MB-231, HCC1937, Panc-1 and Mia PaCa-2 cells were used for evaluation of anticancer properties of synthesized compounds. **Results**: **C-19** showed potent anti-cancer properties, especially enhancement of apoptosis and cell cycle modulation in pancreatic cancer cell lines. In addition, **C-19** and **C-22** showed potent anti-migratory and anti-invasive effects comparable to the well-known ROCK inhibitor fasudil. **Conclusions**: In light of the results of this study, we propose a novel multi-target approach focusing on developing dual HDAC/ROCK inhibitors based on the structure of both **C-19** and **C-22**, exploiting the synergistic potential of these two signaling pathways to improve therapeutic efficacy in metastatic tumors. Our results emphasize the potential of multi-target ROCK inhibitors as a basis for future cancer therapies.

## 1. Introduction

Alterations in the actin cytoskeleton are closely associated with the development and progression of various tumors [1,2]. Changes in the actin cytoskeleton influence cellular processes such as adhesion, contractility, migration, and invasion and contribute to tumor progression through mechanisms such as epithelial-mesenchymal transition (EMT), mesenchymal-epithelial transition (MET), metastasis, neoangiogenesis, extracellular matrix remodeling, vesicular trafficking, and immune cell infiltration. In addition, remodeling of the actin cytoskeleton may influence tumor development by modulating gene expression, cell cycle, and apoptosis [1,3,4]. Overall, actin cytoskeletal remodeling has an impact on cancer cells and cells of the tumor microenvironment, such as endothelial cells and fibroblasts [1].

Rho-associated coiled-coil-containing protein kinases (ROCKs), effectors of the Rho GTPase, are important regulators of the actin cytoskeleton [1,5,6,7]. Two different isoforms belong to this family, ROCK1 and ROCK2, which share 92% similarity in the kinase region and 65% similarity in the overall structure [6]. Activation of ROCKs is mainly mediated by Rho-GTPases. Rho-GTPases switch between an inactive, GDP-bound state and an active, GTP-bound state. Guanine nucleotide exchange factors (GEFs) play a crucial role in activating Rho-GTPases by catalyzing the exchange of GDP for GTP. Conversely, GTPase-activating proteins (GAPs) stimulate the hydrolysis of GTP to GDPGTP hydrolysis to GDP, thereby inactivating Rho-GTPases. Upon activation, Rho-GTPases interact with the Rho-binding domain of ROCKs, inducing a conformational change that exposes the kinase domain of ROCK. This exposure allows the kinase domain to interact with various downstream effectors or molecules, enabling a range of cellular processes (Figure 1). ROCKs can also be activated in a Rho-independent manner by caspase 3 or granzyme B protease cleavage [8,9].

ROCKs influence actin stabilization and actin-myosin contractility by modulating various target proteins, including myosin light chain II (MLC II), myosin phosphatase target subunit 1 (MYPT1), LIM kinases (Figure 1). ROCKs inhibit myosin phosphatase activity by phosphorylating myosin phosphatase target subunit 1 (MYPT1). In addition, ROCKs directly phosphorylate myosin light chain II. Taken together, these alterations lead to the activation of myosin II, which further promotes actin-myosin filament bundling and myosin-driven contraction (Figure 1) [10]. Furthermore, ROCKs phosphorylate and activate LIM kinases, which in turn phosphorylate and inactivate cofilin, an actin-depolymerizing factor (Figure 1). Consequently, the depolymerization of actin filaments is reduced [11]. Overall, these ROCK kinase-induced changes in the activities of MLC II, MYPT1, and LIM kinases lead to a reorganization of the actin cytoskeleton and enhanced myosin-driven contraction. Consequently, tumor progression is promoted, and it can also influence tumor development.

ROCKs are overexpressed in a variety of malignancies, including acute myeloid leukemia, esophageal carcinoma, clear cell renal carcinoma, gastric adenocarcinoma, breast cancer, and pancreatic carcinoma [12]. Although four ROCK inhibitors—fasudil, ripasudil, netarsudil, and belumosudil (Figure 2)—are currently approved for clinical use, none of them have been approved for cancer therapy. Almost all ROCK inhibitors developed to date are Type 1 ATP-competitive kinase inhibitors. A critical component of their structure is the hinge-binding moiety, which is essential for effective ROCK inhibition. Various scaffolds have been used as hinge-binding moieties, including isoquinoline, pyridine, indazole, pyrimidine, pyrrolopyridine, pyrazole, benzamide, aminofurazan, and benzoxaborole, among others [5].

Although most known ROCK inhibitors target both isoforms, the specific roles of ROCK1 and ROCK2 are not yet fully understood. A comprehensive understanding of their individual contributions to tumor development and progression is still lacking. However, some studies have investigated the relationship between isoform expression and cell motility. Shi et al. showed that ROCK1 contributes to destabilization of the actin cytoskeleton by regulating phosphorylation of myosin light chain II (MLC II), whereas ROCK2 affects the actin cytoskeleton primarily through phosphorylation of cofilin [13]. Newell-Litwa et al. demonstrated that ROCK1 is essential for the formation of stable actomyosin filament bundles that contribute to the front-back polarity of cells, while ROCK2 regulates contractile force at the leading edge of migratory cells [14]. Consequently, both isoforms are involved in cell motility and may influence tumor progression and metastasis. However, the role of ROCK1 and ROCK2 in the development and progression of various tumor types may differ, highlighting the need for further research to clarify their specific functions.

Despite extensive preclinical studies demonstrating the anticancer effects of ROCK inhibitors, clinical translation is limited [15,16]. As shown in several preclinical studies, ROCK inhibitors will probably have greater potential in clinical oncology when used as part of combination therapies rather than as monotherapies. For example, in a genetically engineered mouse model of pancreatic adenocarcinoma, the addition of ROCK inhibitors to gemcitabine therapy significantly prolonged survival compared to gemcitabine alone [17]. This improved therapeutic outcome is likely due to the improved uptake of gemcitabine in vivo, facilitated by concomitant treatment with fasudil. In addition, early administration of ROCK inhibitors to pancreatic cancer cells has been shown to inhibit invasion and migration, thereby reducing the metastatic potential and improving the efficacy of chemotherapy [18,19]. In breast cancer models, ROCK inhibitors have been observed to downregulate programmed death-ligand 1 (PD-L1) expression, promoting T-cell activation both in vitro and in vivo. This finding suggests that combining ROCK inhibitors with PD-1/PD-L1 checkpoint blockade could enhance the immune response and represent a promising therapeutic approach for breast cancer [20]. Furthermore, synergistic effects have been demonstrated by in silico and in vitro studies in pancreatic ductal adenocarcinoma when ROCK inhibitors are combined with histone deacetylase (HDAC) inhibitors [21]. Finally, simultaneous inhibition of epidermal growth factor receptor (EGFR) and ROCK has been shown to be a promising strategy for treating triple-negative breast cancer, as revealed by shRNA screening [22]. In drug development today, however, preference is increasingly being given to the development of multitarget-directed ligand strategy over conventional drug combinations. This strategy is the focus of research, as single drugs that can act on multiple targets simultaneously offer significant advantages, particularly in cancer treatment where challenges such as chemoresistance, drug-drug interactions, and poor safety profiles are prevalent [23]. Drugs that act on multiple targets have the potential to effectively address these issues. As far as multi-target ROCK inhibitors are concerned, only one dual ROCK/Akt inhibitor has been identified. This dual inhibitor is the only ROCK inhibitor tested in clinical trials for the treatment of advanced solid tumors. However, the trial results were not positive, mainly due to the drug’s suboptimal pharmacokinetic profile and narrow therapeutic index [24]. Based on the above findings from preclinical, in vitro and in silico studies, numerous combinations of targets can be used for further development of multi-target ROCK inhibitors, which can be a cornerstone in further tumor treatment.

Based on the results of the study of Djokovic N. et al. (2023) [21], one of the key targets that can be used for the design of multi-target ROCK inhibitors are the histone deacetylases (HDAC) that represent crucial regulators of epigenetic processes [21,25]. The HDAC family comprises 18 different isoforms divided into four classes. Although several pan-HDAC inhibitors have been approved for clinical use in tumor treatment, their therapeutic utility is limited by a poor safety profile, mainly due to their lack of selectivity, and their efficacy as monotherapies has been suboptimal [26,27]. Therefore, the development of HDAC inhibitors with selectivity for specific HDAC isoforms and the exploration of multi-target approaches should be prioritized to improve both safety and therapeutic efficacy. Preclinical studies have shown that knockout mice lacking either HDAC4 (belonging to the IIa class of HDAC) or HDAC6 (belonging to the IIb class of HDAC) remain viable and exhibit minimal adverse phenotypes, suggesting that selective inhibition of these isoforms may be well tolerated in vivo [28,29]. Given their specific roles in tumor development and progression, as well as their unique structural characteristics, HDAC4 and HDAC6 have been identified as promising targets for the development of multi-target HDAC/ROCK inhibitors [30,31].

Some studies have shown that the signaling pathways of ROCK and histone deacetylase 6 (HDAC6) intersect during tumor progression, especially considering the cancer cell motility. ROCKs phosphorylate the Tubulin Polymerization Promoting Protein (TPPP1), which leads to the inactivation of TPPP1 and impairs its interaction with HDAC6. As a result, HDAC6 is activated [32]. Therefore, inhibition of ROCK leads to increased tubulin acetylation, while actin cytoskeletal remodeling is also impaired—both essential processes for cancer cell movement. The dual inhibition of ROCK and HDAC6 can, therefore, achieve two effects: stabilization of the microtubule network by enhancing tubulin acetylation and prevention of cancer cell migration by disrupting actin cytoskeleton dynamics and stabilizing microtubule dynamics. In this way, one of the possible synergistic mechanisms between HDAC and ROCK inhibitors was revealed. However, further studies are needed to fully understand the synergistic mechanisms. All in all, we can assume that the integration of HDAC and ROCK inhibitory functions in a single molecule could potentially increase the efficacy of anticancer drugs by simultaneously targeting multiple signaling pathways involved in cancer cell proliferation, survival, and metastasis, thus representing a new and effective strategy for cancer therapy.

Building on the prior findings and the therapeutic potential of ROCK inhibitors, the primary objectives of this study are as follows:(1)To identify key structural features of ROCK1/ROCK2 inhibitors using computer-aided drug design methods, especially ligand-based approaches such as 3D-Quantitative Structure-Activity Relationship (3D-QSAR) and structure-based methods such as molecular docking;(2)To develop novel ROCK inhibitors and synthesize compounds with the best-predicted activity based on the computational models;(3)To evaluate the activity of the synthesized ROCK inhibitors using enzyme and cancer cell assays to assess their efficacy and potential as therapeutic agents;(4)To propose future perspectives on multi-target HDAC/ROCK drug design based on the structure of novel developed ROCK inhibitors.

## 2. Materials and Methods

### 2.1. Design of Novel ROCK Inhibitors

#### 2.1.1. Preparation of Compounds for Molecular Docking and 3D-QSAR Study

The 49 ROCK1 and 47 ROCK2 inhibitors were selected for this study [33,34,35]. Based on their structure, they were divided into four clusters: I cluster—isoquinoline derivatives, II cluster—pyridine derivatives, III cluster—indazole derivatives, and IV cluster—pyrazole derivatives. The structures of these compounds with their IC_50_ values were obtained from the ChEMBL database [36].

Initially, all compounds were prepared for this study using the program Marvin Sketch 6.1.0 (Chem Axon 2013) [37], which selected all dominant microspecies at a physiological pH = 7.4. The dominant microspecies were further optimized by applying the semi-empirical PM3 [38] and Hartree-Fock with the 3-21G basis set method [39], available in Gaussian 09 software [40] as a part of the ChemBio3D Ultra 13.0 program [41].

#### 2.1.2. Molecular Docking

The molecular docking study was performed using Gold Software 2022.1.0 [42] in order to visualize and study the key interactions between enzymes and compounds, rank molecules according to their affinity for ROCK1 and ROCK2, and generate bioactive conformations of all ligands for the 3D-QSAR study. The crystal structures of ROCK1 (PDB: 6E9W), ROCK2 (PDB: 7JNT), and HDAC6 (PDB:6EDU) were downloaded from the Protein Data Bank [43]. The three-dimensional (3D) structures were prepared for molecular docking using the online software Play Molecule—ProteinPrepare [44], which removed all non-bound water and heteroatoms and added all missing hydrogen atoms.

The virtual docking procedures for both enzymes were validated for further predictions by calculating root-mean-square deviation (RMSD) values. Ligands were ranked based on their affinity for the enzymes by using the ChemPLP scoring function (ROCK1 and ROCK2) and the Chem scoring function (HDAC6). The observed interactions between all molecules and ROCK1 and ROCK2 were visualized by the Discovery Studio 2020 program [45] and PyMOL 3.0.4.

#### 2.1.3. Development of 3D-QSAR (ROCK1) and 3D-QSAR (ROCK2) Models

3D-QSAR (ROCK1) and 3D-QSAR (ROCK2) models were developed by using Pentacle software 1.07. The main purpose of these models was to identify critical structural features required for effective inhibition of ROCK1 and ROCK2 kinases.

The development of 3D-QSAR (ROCK1) and 3D-QSAR (ROCK2) models was based on the calculation of alignment-independent three-dimensional molecular descriptors (GRIND) [46]. The concept of computing GRIND variables based on the interactions between four different chemical probes and examined ligands is known as the Molecular Interaction Fields (MIF) methodology [47]. The probes used to mimic ligand–target interactions are the TIP probe, O probe, N1 probe, and DRY probe. Hydrophobic interactions are described by the DRY probe, while the TIP probe is used to evaluate steric interactions and compound shape. The O probe (carbonyl-oxygen) and the N1 probe (amide-nitrogen) are used to mimic hydrogen bonding interactions. Grid spacing is set to 0.5 Å by default. The extraction of the main MIFs was performed using the algorithm ALMOND and it was based on the distance between the nodes and their intensity [47]. The Consistently Large Auto and Cross Correlation (CLACC) algorithm calculated the GRIND variables with a smoothing window set to 0.8 Å.

The most significant GRIND variables for partial least squares (PLS) regression were selected using fractional factorial design (FFD) [48].

3D-QSAR (ROCK1) and 3D-QSAR (ROCK2) models were constructed using the negative logarithm of the IC_50_ values as dependent variables, whereas the GRIND molecular descriptors were used as independent variables. Finally, PLS regression was performed by Pentacle software to generate the 3D-QSAR (ROCK1) and 3D-QSAR (ROCK2) models.

#### 2.1.4. Selection of Training and Test Set Compounds

All compounds were divided into training (34 compounds for 3D-QSAR (ROCK1) and 32 compounds for 3D-QSAR (ROCK2)) and test sets (15 compounds for 3D-QSAR (ROCK1) and 15 compounds for 3D-QSAR (ROCK2)) to ensure an even distribution of the entire pIC_50_ range across both datasets (Appendix A). The compounds from each cluster were evenly distributed across the training and test sets for both models. Prior to partitioning, a Principal Component Analysis (PCA) of the dataset was performed using Pentacle 1.0.7 software to assess and validate the distribution and clustering of the compounds.

#### 2.1.5. Internal and External Validation of Model

In order to validate formed 3D-QSAR (ROCK1) and 3D-QSAR (ROCK2) models, the following parameters are calculated: R^2^ (coefficient of determination) (Equation (1)), PRESS (Predicted Residual Sum of Squares) (Equation (2)), Q^2^_LOO_ (Leave-One-Out Cross-Validated squared correlation coefficient) (Equation (3)), and RMSEE (Root Mean Square Error of Estimation) (Equation (4)) [49,50].

R^2^ is calculated for the compounds of the training set using the observed pIC_50_ values (Y_training_), the predicted values pIC_50_ (Ŷ_training_), and the average Y-value (Ȳ_training_) (Equation (1)). The Q^2^_LOO_ (Equation (3)) is performed by the leave-one-out (LOO) technique, where each compound is removed from the training set, and then a new model is created using the remaining molecules in the training set. The new model is used to predict the Y-value of the removed compound. This process is repeated until all ligands from the training set have been removed once.

The following equations are used to examine the internal predictive ability and robustness of the developed models (Equations (1)–(4)):(1)$$R^{2}=\frac{\sum {\left({Ŷ}_{training}-{Ȳ}_{training}\right)}^{2}}{\sum {\left(Y_{training}-{Ȳ}_{training}\right)}^{2}}$$
(2)$$PRESS=\sum_{i-1}^{n}e_{\left(i\right)}^{2}$$
(3)$$Q^{2}=1-\frac{PRESS}{\sum {\left(Y_{training}-{Ȳ}_{training}\right)}^{2}}$$
(4)$$RMSEE=\sqrt{\frac{PRESS}{n}}$$

The ability of the 3D-QSAR (ROCK1) and 3D-QSAR (ROCK2) models to predict the bioactivity of the molecules from the training set is estimated by the cross-validated Q^2^_LOO_ value, which should be above 0.5.

External validation of 3D-QSAR (ROCK1) and 3D-QSAR (ROCK2) models was performed using test set molecules. The following parameters are used for external validation: R^2^_pred,_ RMSEP (root mean square error of prediction), and r^2^_m_ metrics [51]. The developed 3D-QSAR (ROCK1) and 3D-QSAR (ROCK2) models have good predictive properties when the values of $r_{m}^{2}$ and r^/2^_m_ are close to each other and higher than 0.5. The difference between these two values (Δr^2^_m_) should be less than 0.2, while R^2^_pred_ should be greater than 0.5. These parameters are calculated using the following equations (Equations (5)–(8)):(5)$$R_{pred}^{2}=1-\frac{\sum {\left(Y_{test}-{Ŷ}_{test}\right)}^{2}}{\sum {\left(Y_{test}-{Ȳ}_{training}\right)}^{2}}$$
(6)$$RMSEP=\sqrt{\frac{PRESS}{n}}$$
(7)$$r_{m}^{2}=r^{2}(1-\sqrt{r^{2}-r_{0}^{2}})$$
(8)$$r_{m}^{/2}=r^{2}(1-\sqrt{r^{2}-r_{0}^{/2}})$$

#### 2.1.6. Applicability Domain

The applicability domain (AD) represents the structural, physicochemical, or biological space in which the developed model can actually predict the activity of a test set or novel design compounds. Therefore, the prediction of activity using QSAR is valid only if the novel design compounds fall within the AD of the model [52]. The AD of QSAR is defined by the properties of the ligands from the training and test dataset. The applicability domain for 3D-QSAR (ROCK1) and 3D-QSAR (ROCK2) models was determined by a leverage approach performed using SPSS v.18.0 software. Williams plots were created, and the critical leverage h* (the vertical line) is calculated using the following equation (Equation (9)):h* = 3(p + 1)/n(9)
where n represents the number of compounds in the training set, while p represents the key GRIND variables used to define AD. If the absolute standardized residual for a compound is more than three standard deviation units and/or the leverage value of a compound is above the critical value (h*), the compound is outside the AD of the model, and the models formed cannot be used for reliable prediction of new molecule activity.

### 2.2. Synthesis—General Procedures

All reagents used in the synthesis were obtained commercially and were of reagent purity or better (e.g., Merck (Darmstadt, Germany), Sigma Aldrich (St. Louis, MO, USA), Fisher Scientific (Hampton, NH, USA)). The ^1^H and ^13^C nuclear magnetic resonance (NMR) spectra were recorded using a Bruker Ascend 400 (400 MHz) spectrometer. Deuterochloroform and deuterated methanol were used as solvents, and chemical shifts were reported in parts per million (δ) relative to tetramethylsilane as an internal reference. Mass spectrometry data were obtained using an LTQ Orbitrap XL. Silica gel 60 (230–400 mesh) was used for flash chromatography, and thin-layer chromatography (TLC) was performed on alumina plates with a 0.25-mm-thick silica gel layer (Kieselgel 60 F254; Merck, Darmstadt, Germany). The compounds were detected by staining with potassium permanganate.

#### 2.2.1. Synthesis of Benzyl Bromide Derivatives—General Procedure A

The benzaldehyde derivative (1.25 eq) was dissolved in 10 mL of 1 M NaOH solution. Sodium borohydride (NaBH_4_-1 eq) was then gradually added to the solution, keeping the temperature at 0 °C. The reaction mixture was stirred for 30 min at room temperature. Subsequently, a 2 M HCl solution was added at 0 °C until neutralization. The resulting benzyl alcohol was extracted with ethyl acetate, which was then dried over anhydrous Na_2_SO_4_. After the ethyl acetate had evaporated, the remaining oil was dissolved in dichloromethane (DCM) 10 mL. Phosphorus tribromide (PBr_3_, 1.5 eq) was added dropwise while maintaining the temperature at 0 °C. The reaction mixture was stirred for 3 h at room temperature. Following the reaction, an ice-cold solution of sodium bicarbonate (NaHCO_3_) was added to neutralize the mixture. The resulting benzyl bromide derivatives were extracted with DCM, which was subsequently dried and evaporated. The remaining oil was then used for the final step of the synthesis.

#### 2.2.2. Synthesis of Homopiperazine Derivatives—General Procedure B

To a solution of corresponding acid (500 mg) in dimethylformamide (DMF) (0.5 mL), thionyl chloride (3 mL) was carefully added, and the reaction mixture was heated to 80 °C overnight. After cooling to room temperature, the thionyl chloride and DMF were removed under reduced pressure. The remaining oil was then dissolved in acetonitrile (5 mL), to which 1-Boc-homopiperazine (1 eq) and triethylamine (3 eq) were added. The reaction mixture was heated to reflux for 3 h. After cooling, the acetonitrile was evaporated, and the residual oil was dissolved in dichloromethane (5 mL). Trifluoroacetic acid (0.7 mL) was added and the mixture was stirred overnight at room temperature, monitoring the progress of the reaction by TLC. After evaporation of the volatiles, a saturated sodium bicarbonate solution was added, and final compounds—amine derivatives—were extracted with ethyl acetate (3 × 10 mL). The ethyl acetate extracts were dried over anhydrous Na_2_SO_4_ and evaporated. The structure of products was confirmed by 1H NMR, and compounds were used for further synthesis without purification.

#### 2.2.3. Synthesis of Novel Developed ROCK Inhibitors—General Procedure C

A benzyl bromide derivative (1.1 eq) and triethylamine (2 eq) were added to a solution of the corresponding amine (1 eq) in acetonitrile (10 mL). The reaction mixture was then refluxed overnight. After completion, the acetonitrile was evaporated under reduced pressure. The residue was neutralized with an aqueous solution of NaHCO_3_, and the resulting mixture was extracted with dichloromethane (DCM). The organic layer was dried over anhydrous Na_2_SO_4_, and after filtration solvent was removed under reduced pressure. The crude mixture was purified by flash chromatography using a gradient—100% DCM, then DCM:MeOH (90:10), yielding the final product—a novel ROCK inhibitor.

### 2.3. ROCK Enzyme Assays

All synthesized ROCK inhibitors were evaluated against ROCK1 and ROCK2 using a 10-dose IC_50_ method with a 3-fold serial dilution starting at 100 µM. The control compound, staurosporine, was tested using a 10-dose IC_50_ method with a 4-fold serial dilution starting at 20 µM. ROCK kinases (ROCK1 and ROCK2) and the Long S6 Kinase substrate peptide were prepared in a Reaction Buffer, which contained 20 mM Hepes (pH 7.5), 10 mM MgCl_2_, 1 mM EGTA, 0.01% Brij35, 0.02 mg/mL BSA, 0.1 mM Na_3_VO_4_, 2 mM DTT, and 1% DMSO. The compounds were added to the reaction and incubated at room temperature for 20 min. Subsequently, 33P-ATP was added to the reaction mixture (10 µM for ROCK1 and 50 µM for ROCK2). The prepared mixture was stirred for 2 h at room temperature, and ROCK kinase activity was detected using the P81 filter binding method [53]. ROCK enzyme assays were performed by the Reaction Biology Corporation.

### 2.4. Biological Evaluation

In vitro anticancer potential was evaluated on a panel of four cell lines (2 breast cancer and 2 pancreatic cancer cells). Triple negative breast cancer cells, BRCA1-competent (MDA-MB-231, ATCC^®^, HTB-26™) and BRCA1-mutated (HCC1937, ATCC^®^, CRL-2336™) were used. The cells were maintained in monolayer cultures: MDA-MB-231 in DMEM:Ham’s 12 (1:1) medium (Sigma-Aldrich, St. Louis, MO, USA) and HCC1937 in RPMI (Sigma-Aldrich, St. Louis, MO, USA) supplemented with penicillin (192 U/mL), streptomycin (200 µg/mL) and 10% heat-inactivated fetal calf serum (FCS). Both pancreatic cancer cell lines have KRAS and TP53 mutations. Mia Paca-2 (ATCC^®^, CRL-1420) and Panc-1 (ATCC^®^, CRL-1469) cells were cultured in DMEM medium (Sigma-Aldrich, St. Louis, MO, USA) supplemented with 10% fetal bovine serum (FBS) (Sigma-Aldrich, St. Louis, MO, USA), 100 µg/mL streptomycin and 100 UmL-1 penicillin (Sigma-Aldrich, St. Louis, MO, USA). All cells were grown as monolayers in a humidified atmosphere of 95% air and 5% CO_2_ at 37 °C.

#### 2.4.1. Cell Viability (MTT) Assay

The cytotoxic activity of the synthesized compounds was assessed by colorimetric MTT assay on MDA-MB-231, HCC1937, Mia PaCa-2, and Panc-1 cells and compared with fasudil [54]. Cells were seeded in 96-well microtiter plates at the following densities: MDA-MB-231 (5 × 10^3^ cells/well), HCC 1973 (5 × 10^3^ cells/well), Mia PaCa-2 (2 × 10^3^ cells/well), and Panc-1 (3 × 10^3^ cells/well). After adhesion, the cells were treated 24 h later with the novel synthesized compounds (concentrations 100 nM–3.13 μM). After 72 h of incubation, 10 μL of MTT (3-(4,5-dimethylthiazol-2-yl)-2,5-diphenyltetrazolium bromide) solution (5 mg/mL phosphate-buffered saline) was added. The samples were then incubated for 4 h at 37 °C in a humidified atmosphere with 5% CO_2_. Subsequently, 100 μL of 100 g/L sodium dodecyl sulfate was added. The absorbance was measured 24 h later at 570 nm (Multiskan EX Reader, Thermo Labsystems, Vantaa, Finland). The cell survival rate (%) corresponds to the absorbance ratio of treated and control cells multiplied by 100. The concentration of synthesized ROCK inhibitors that reduces cell viability by 50% is defined as the IC_50_ value and compared with the control. The toxicity of all compounds was evaluated in a normal human fetal lung fibroblast cell line (MRC-5) to determine the selectivity index (Si) of the molecules.

#### 2.4.2. Cell Cycle Analysis

Mia Paca-2 and Panc-1 cells were treated with 20 μM solutions of the tested compounds for 48 h. After treatment, cells were washed in cold PBS and incubated in 96% ethanol on ice for 30 min, centrifuged and incubated with 80 μL RNase A (200 μg/mL/mL) and 50 μL propidium iodide (PI) (50 μg/mL) at 37 °C for 30 min. Flow cytometry (FACS Calibur E440, Becton Dickinson cytometer, and BD CellQuest Pro software (version 5.2.1.)) was used to assess the percentage of a cell population in the different phases of the cell cycle. The results were presented as the percentage of cells in the different cell cycle phases.

#### 2.4.3. Apoptosis Assay

Apoptosis of treated cells was determined using the BD Pharmingen FITC Annexin V assay kit (BD Pharmingen, San Diego, CA, USA). Mia PaCa-2 and Panc-1 cells were treated with 20 μM solutions of the tested compounds for 48 h and further processed according to the procedure indicated by the manufacturer. After the incubation period, the cells were stained with Annexin V fluorescein isothiocyanate, and PI Apoptosis was monitored by flow cytometry (FACS Calibur E440, Becton Dickinson cytometer, and Cell Quest software).

#### 2.4.4. Detection of Mitochondrial Membrane Potential

The membrane potential of mitochondria was determined using the fluorescent dye JC-1 [55]. Mia PaCa-2 and Panc-1 cells were treated with 20 μM solutions of the tested compounds for 24 h. After treatment, the cells were stained with 40 μL JC-1 (final concentration 15.4 μM) and incubated for 15 min under standard conditions. The fluorescence intensity of the cells was determined by flow cytometry (FACS Calibur E440, Becton Dickinson cytometer, and Cell Quest software). The JC-1 dye accumulates in the mitochondria in a potential-dependent manner, which is indicated by a green fluorescence emission that shifts to red with a concentration-dependent formation of red fluorescent J-aggregates. Mitochondrial depolarization is indicated by a decrease in the ratio between red and green fluorescence intensity.

#### 2.4.5. Evaluation of ROS Production

The levels of total and mitochondrial ROS in Mia PaCa-2 and Panc-1 cells were observed after 24 h exposure to 20 μM solutions of the synthesized compounds. According to the previously described method, cells were harvested after treatment, washed with PBS, and incubated with 2′,7′-dichlorodihydrofluorescein diacetate (30 μM in PBS) for 45 min at 37 °C, then washed again and analyzed. MitoSOX Red was used to detect superoxide in the mitochondria of living cells. After an incubation period of 48 h, the cells were harvested, washed twice, resuspended in 5 μM MitoSOX, and incubated for 10 min at 37 °C in the dark. The levels of intracellular ROS and mitochondrial superoxide were analyzed by flow cytometry (FACS Calibur, BD Biosciences, San Diego, CA, USA). The fluorescence intensity emitted by dichlorofluorescein and MitoSOX was determined by flow cytometry (FACS Calibur E440, Becton Dickinson cytometer, and Cell Quest software).

#### 2.4.6. Generation and Analysis of Tumor Spheroids

MDA-MB-231, Mia PaCa-2, and Panc-1 cells were seeded at a cell density of 1000 c/w in 100 μL DMEM with 10% FCS in a NunclonSphera 96-well ultra-low attachment plate (Thermo Scientific Nunc™) to form multicellular tumor spheroids for 4 days. The formation and growth of tumor spheroids were examined and imaged using an Olympus CKX53 microscope with a 4× objective. Breast and PDAC tumor spheroids were treated by carefully adding 50 μL of the medium with fresh nutrient medium for control spheroids or with a compound-supplemented medium at the indicated concentrations for treated spheroids for an additional 72 h. The cytotoxicity of the drugs towards the tumor spheroids was investigated using the MTT assay.

In another experimental setup, breast and pancreatic cancer cells were seeded in co-culture with the tested compounds at the indicated concentrations and a cell density of 1000 c/w in 150 μL DMEM containing 10% FCS in a NunclonSphera 96-well, ultra-low attachment plate (Thermo Scientific Nunc) to form tumor spheroids for 4 days. The formation and growth of tumor spheroids were examined and imaged using an Olympus CKX53 microscope with a 4× objective. The cytotoxicity of the compounds against the breast and pancreatic tumor spheroids was examined using the MTT assay.

#### 2.4.7. Wound Healing Assay

To determine the HCC1973 and Panc-1 cells’ migration capabilities, 1 × 10^5^ cells were seeded in 24-well plates and cultured until they formed a confluent monolayer. Then, the cell monolayers were scratched with a 200-μL pipette tip, washed three times with PBS to remove unattached cells, and cultured for another 24 h under the specified treatments. At the end of the treatment period, the culture medium was removed, and the cells were fixed with ice-cold 100% methanol, washed twice with PBS, and stained with 0.1% crystal violet for 30 min. Wound closure was documented using an inverted light microscope and quantified using NIH ImageJ software (NIH image program, version 1.49).

#### 2.4.8. Transwell Migration and Invasion Assays

To determine the migration and invasion ability of MDA-MB-231 and Mia PaCa-2, Corning^®^ Costar^®^ Transwell^®^ cell culture inserts with polycarbonate filters with 8.0 μm pore size (Sigma Aldrich, CLS3464) and Corning™ BioCoat™ Matrigel™ Invasion Chamber with Corning Matrigel Matrix™ cell culture inserts (Thermo Fisher Scientific, Waltham, MA, USA) were used. 8 × 10^4^ cells were seeded in the upper chamber in a 200 μL medium. The lower chamber was then filled with an 800 μL medium. After 24 h of specified treatments, the cells remaining in the upper chamber were carefully removed with a cotton swab and then moistened with the medium. After 24 h of defined treatments, the cells remaining in the upper chamber were carefully removed with a cotton swab and then moistened with the medium. The Transwell inserts were carefully washed thrice with 37 °C pre-warmed PBS. Cells adhering to the bottom of the transwell membrane were fixed with ice-cold 100% methanol for 2 min and stained with 0.1% crystal violet for 30 min at room temperature. Invasive and migrating cells were photographed and quantified using NIH ImageJ software (NIH image program, version 1.49).

## 3. Results and Discussion

### 3.1. Design of Novel ROCK Inhibitors

#### 3.1.1. Results of 3D-QSAR Study for ROCK1 and ROCK2

The developed 3D-QSAR (ROCK1) and 3D-QSAR (ROCK2) models are used to determine critical structural features relevant to effective dual ROCK1/ROCK2 inhibition. The results of the internal and external validation of the two models are shown in Table 1 and Table 2. According to the validation parameters, both models are reliable and can be used to predict the ROCK1/ROCK2 inhibiting activity of novel designed molecules. The distribution of experimental pIC_50_ values of the compounds in the training and test set, along with the pIC_50_ values predicted by the created 3D-QSAR (ROCK1) and 3D-QSAR (ROCK2) models, are shown in Appendix A. All compounds (Appendix A) from training and test sets were in the applicability domain of developed 3D-QSAR (ROCK1) and 3D-QSAR (ROCK2) models (Appendix A).

The main variables affecting ROCK1 activity are shown in the PLS coefficient (Appendix A).

The favorable var323 (DRY-N1: 13.2–13.6 Å) and var328 (DRY-N1: 15.2–15.6 Å) are described in all four clusters between the hydrogen bond accepting moiety—the nitrogen or oxygen atom of the heterocycle (tetrahydroisoquinoline, pyridine, indazole or pyrazole) and the hydrophobic spot around the phenyl group and linker. These variables are not present in the less active molecules such as CHEMBL67352 (pIC_50_(ROCK1) = 4.120), CHEMBL20232147 (pIC_50_(ROCK1) = 4.230) and fasudile (pIC_50_(ROCK1) = 6.180), while it is observed in the most active compounds such as CHEMBL1922035 (pIC_50_(ROCK1) = 8.700) and CHEMBL1922128 (pIC_50_(ROCK1) = 8.520—Figure 3A). CHEMBL20232147 (pIC_50_(ROCK1) = 4.230) does not have the bulky substituents on the indazole, including the phenyl group, as well as CHEMBL67352 (pIC_50_(ROCK1) = 4.120) and fasudil (pIC_50_(ROCK1) = 6.180). Thus, we can conclude that the presence of a heterocyclic moiety with a hydrogen bond-accepting group located at an optimal distance from the aromatic substituents, such as the phenyl group, is very important for the inhibition of ROCK1.

The favorable var573 (N1-TIP: 20.4–20.8 Å) is observed for all molecules having a substituent on the phenyl ring in meta or para position, including the most active ones (CHEMBL1922035 (pIC_50_(ROCK1) = 8.700), CHEMBL1922128(pIC_50_(ROCK1) = 8.520, Figure 3A). It is represented by the optimal distance between the hydrogen bond accepting group—nitrogen atom of the heterocycle and the steric spot around the methyl group or the hydroxyl group of the phenyl ring. This variable is not present in the least active compounds such as CHEMBL2023148 (pIC_50_(ROCK1) = 3.800), CHEMBL67352 (pIC_50_(ROCK1) = 4.120) and CHEMBL2023147 (pIC_50_(ROCK1) = 4.230). In addition, the importance of the substituents of the phenyl ring is shown by favorable var383 (DRY-TIP: 14.0–14.4), highlighting the positive influence of the distance between the hydrophobic region around the heterocycle (pyridine or indole) and the steric spot around the substituents of benzene on the ROCK1 inhibitory activity. Consequently, we can assume that the increase in ROCK1 inhibitor activity is due to the presence of meta- or para-substituents.

Moreover, the introduction of hydrogen bond-accepting groups correlates positively with the inhibition of ROCK1, as confirmed by var164 (N1-N1: 19.2–19.6 Å) observed between the hydrogen bond-accepting group of the heterocycle (pyrazole, indazole, or pyridine) and the hydrogen bond-accepting group of the phenyl ring. This variable var164 is found in the most active molecules of the training set (Figure 3A).

Regarding the linker (connecting two parts of the molecules—usually a heterocycle and a phenyl group), the unfavorable var64 (O–O: 2.4–2.8 Å) underlines the negative influence of the ureido group on the activity of ROCK1 inhibitors. It is usually observed between hydrogen bond-donating groups—the nitrogen atoms of the ureido group. In some molecules (CHEMBL1922143, CHEMBL1922142, and CHEMBL1922034), var64 is described between the unsubstituted nitrogen atom of the indole and the nitrogen atom of the carbamoyl group, while it is absent in the derivatives of the cluster IV, whose nitrogen atoms are substituted with different alkyl groups (CHEMBL1922042, CHEMBL1922043, CHEMBL1922135, CHEMBL1922044, CHEMBL1922134, CHEMBL1922133, CHEMBL1922030, CHEMBL1922126, CHEMBL1922129, CHEMBL1922136, CHEMBL1922035, CHEMBL1922127, CHEMBL1922033, CHEMBL1922128, CHEMBL1922125). Based on these findings, we can conclude that the introducing substituent on the nitrogen atoms of the ureido group should be considered in order to eliminate the negative effects of var64 and increase the activity of ROCK1 inhibitors.

The key variables correlating with ROCK2 activity are shown by the PLS coefficient plot (Appendix A).

The favorable var254 (DRY-O: 2.4–2.8 Å) indicates the importance of an optimal distance between the hydrophobic region around the heterocycle and the hydrogen bond-donating group—unsubstituted nitrogen atom of the linker. In addition, **var459** (O-N1: 10.0–10.4 Å) underscored the impact of a specific distance between the hydrogen bond-donating group of the linker and the hydrogen bond-accepting group of the heterocycle (Figure 3B). These variables suggest that the heterocycle with a hydrogen bond-accepting group (nitrogen atom) that is linked to the hydrogen bond-donating group in the linker (unsubstituted nitrogen atom) is a critical structural feature for ROCK2 inhibition.

The presence of the favorable var355 (DRY-N1: 18.0–18.4 Å) suggests that the distance between the hydrogen bond-accepting group of the heterocycle (nitrogen atom) and the hydrophobic region of the phenyl ring positively correlates with the activity of ROCK2 inhibitors. It is present in the most active inhibitors, while it is not described in the least active molecule CHEMBL2023148 (pIC_50_(ROCK2) = 4.210), probably due to the insufficient length of the linker between the heterocycle and the phenyl group. Based on these results, we can assume that the presence of an aromatic ring, such as the phenyl group, can increase the activity of the ROCK2 inhibitors only if this moiety is located at an appropriate distance—0.4 Å from the heterocycle.

The favorable var161 (N1-N1: 14.8–15.2 Å) underlined that the length of the linker is of crucial importance and should be considered in detail in the development of novel inhibitors. This variable described the optimal distance between the hydrogen bond-accepting group of the heterocycle (pyridine, pyrazole, isoquinoline) and the hydrogen bond-accepting group of the linker (oxygen atom of the carbonyl group). This is not the case for some compounds of cluster III, such as the least active CHEMBL2023148 (pIC_50_(ROCK2) = 4.210), even though the hydrogen-accepting group of the linker is present. From these results, we can conclude that we should consider not only the presence of the hydrogen bond-accepting group but also the optimal distance of this group—0.4 Å from the hydrogen-accepting group of the heterocycle in order to design more effective ROCK2 inhibitors. The distance between the hydrogen bond accepting group of the heterocycle (pyridine, pyrazole, isoquinoline, or indazole) and the hot spot around the substituents of the phenyl ring (in para or meta position) is established as significant for the inhibition of ROCK2 by var597 (N1-TIP: 15.60–16.0 Å) (Figure 3B). Moreover, the introduction of the hydrogen bond-accepting group as substituents of the phenyl ring is described as important by var477 (O-N1: 17.2–17.6 Å), which was observed between the hydrogen bond-donating groups of the heterocycle (nitrogen or oxygen atoms) and the hydrogen bond accepting groups—the substituents in meta or para position of the phenyl ring (Figure 3B). Thus, the introduction of hydrogen bond-accepting groups into the phenyl ring will clearly correlate with the effective ROCK2 inhibition.

Considering all previous results, we can assume that the following structural features are necessary for dual ROCK1/ROCK2 inhibition:(1)Heterocycle with nitrogen atom as hydrogen-accepting group due to positive impact of var323 (DRY-N1: 13.2–13.6 Å) (ROCK1), var328 (DRY-N1: 15.2–15.6 Å) (ROCK1), var254 (DRY-O: 2.4–2.8 Å) (ROCK2), var459 (O-N1: 10.0–10.4 Å) (ROCK2);(2)Linker with hydrogen accepting and hydrogen-donating group (amide group), which is shown by var459 (O-N1: 10.0–10.4 Å) (ROCK2);(3)Phenyl ring with hydrogen bond accepting groups as substituents in *meta* or *para* position due to the influence of var573 (N1-TIP: 20.4–20.8 Å) (ROCK1), var383 (DRY-TIP: 14.0–14.4 Å) (ROCK1), var164 (N1-N1: 19.2–19.6 Å) (ROCK1) and var597 (N1-TIP: 15.60–16.0 Å) (ROCK2), var477 (O-N1: 17.2–17.6 Å) (ROCK2).

#### 3.1.2. Molecular Docking Study

Both docking procedures performed with ROCK1 (PDB:6E9W) and ROCK2 (PDB: 7JNT) enzymes can be considered valid, which is confirmed by calculating the RMSD values for each enzyme.

The nitrogen atom of the pyridine ring in the co-crystal ligand J0P501 (ChemPLP Score: 82.68, PDB: 6E9W) forms a key interaction with Met156, which contributes significantly to ROCK1 inhibition. In addition, the J0P501–ROCK1 complex is stabilized by hydrogen bonds, one between the carbonyl group of the ligand and Lys105 and another between the oxygen atom of 1,4-dioxane and Phe87. Numerous hydrophobic interactions with residues such as Leu205, Ala103, Val90, Ala215, Arg84, Met153, and Gly85 further facilitate the anchoring of J0P501 to the ROCK1 binding site (Appendix A).

The hinge-binding pyrazole moiety of the most active compound, ChEMBL1922035 (pIC_50_ = 8.70, ChemPLP Score: 83.36, cluster IV—Figure 4A), forms essential hydrogen bonds for optimal binding within the enzyme’s active site with Met 156 and Glu154. These key interactions (both or one of them) are consistently observed between co-crystal ligand J0P501 (Appendix A) as well as in other potent inhibitors across various clusters, such as RKI1447 (pIC_50_ = 8.10, ChemPLP Score: 78.68, cluster II—Appendix A), ChEMBL1084892 (pIC_50_ = 7.85, ChemPLP Score: 68.26, cluster I—Appendix A), and ChEMBL2023160 (pIC_50_ = 5.85, ChemPLP Score: 64.36 cluster III—Appendix A). Conversely, the least active compound in the training set, CHEMBL2023148 (pIC_50_ = 3.80, ChemPLP Score: 60.97—Figure 4B), is notably devoid of hydrogen bonds with Glu154 and Met156. This correlation suggests that the formation of hydrogen bonds between the hinge-binding moiety and Glu154 and/or Met156 is a critical determinant for effective ROCK1 inhibition.

In addition, the oxygen atom of the carbonyl linker and the methoxy group on the phenyl ring of ChEMBL1922035 form hydrogen bonds with Lys105 and Phe87, respectively, enhancing the anchoring of the compound in the enzyme (Figure 4A), consistent with the co-crystal ligand J0P501. Remarkably, hydrogen bonding with Phe87 is also present in other highly active inhibitors with a meta-substituted phenyl ring, such as RKI1447 (pIC_50_ = 8.10—Appendix A), CHEMBL1922140 (pIC_50_ = 8.22), CHEMBL1922127 (pIC_50_ = 8.05) and CHEMBL1922128 (pIC_50_ = 8.52). In contrast, this interaction is absent in the least active compounds (CHEMBL2023148–pIC_50_ = 3.80—Figure 4B; CHEMBL67352–pIC_50_ = 4.12—Appendix A), which lack a phenyl ring substituted by a hydrogen-donating group, indicating a positive correlation between such substitutions and ROCK1 inhibition.

Finally, numerous van der Waals and π-sigma interactions are observed between the pyrazole, 7-azaindole, isoquinoline, or phenyl ring and various amino acid residues, including Ala103, Val137, Leu205, Met153, Phe120, Val90, and Gly85, which further stabilize the binding of compounds within the ROCK1 binding site. This interaction pattern is similar to that of the co-crystal ligand J0P501. However, for some less active compounds such as CHEMBL67352 (Appendix A) and CHEMBL2023147 (Appendix A), hydrogen bonds between hinge binding moiety and residues are present, but these molecules do not have good ROCK1 inhibitory activity. The lack of hydrogen bonds and π-sigma interactions with the linker or phenyl ring suggests that these interactions should be carefully considered when designing new ROCK1 inhibitors.

The co-crystal ligand of ROCK2-VFA501 (ChemPLP Score: 82.46, PDB:7JNT) formed a hydrogen bond with Met172, which is considered crucial for ROCK2 inhibition (Appendix A). In addition, two hydrogen bonds are formed between VFA501 and ROCK2, one between the carbonyl group of the linker and Lys121 and another between the methoxy group and Phe103. Finally, numerous van der Waals interactions are described between VFA501 and various amino acid residues such as Val106, Val153, Met169, Ala119, Leu221, Arg100, Phe136, Ala102, etc., which positively correlate with ROCK2 inhibition (Appendix A).

The most active compound in the ROCK2 training set, ChEMBL1922128, which belongs to cluster IV (pIC_50_= 9.00, ChemPLP Score: 93.01), forms two critical hydrogen bonds between the NH group of the pyrazole hinge-binding moiety and the amino acid residues Met172 and Glu170 that is consistent with the binding mode of co-crystal ligand VFA501 (Figure 5A and Appendix A). The hydrogen bonds between hinge-binding moiety—nitrogen atoms of isoquinoline, pyridine, or indazole and Met172 or/and Glu170 are also present in the most active compounds of other clusters—ChEMBL3689470 (pIC_50_ = 7.130, ChemPLP Score: 76.20 cluster I—Appendix A), RKI1447 (pIC_50_ = 8.210, ChemPLP Score: 79.26, cluster II—Appendix A) and ChEMBL2023154 (pIC_50_ = 6.170, ChemPLP Score: 67.13—Appendix A). However, the least active compound of the training set—ChEMBL2023148 (pIC_50_ = 4.210, ChemPLP Score: 50.26—Figure 5B) did not form a hydrogen bond with Met172 or Glu170 highlighted the importance of the presence of these interactions for ROCK2 inhibitory activity.

The methoxy group at the meta-position of the phenyl ring (ChEMBL1922128-Figure 5A) forms a hydrogen bond with Phe103, an interaction observed in other highly active compounds, such as ChEMBL1922125 (pIC_50_ = 8.30, ChemPLP Score: 90.84—Appendix A) and RKI1447 (pIC_50_ = 8.21—Appendix A) as well as co-crystal ligand VFA501 (Appendix A). Additionally, the methoxy group of the phenyl ring engages in hydrophobic interactions with the P-loop of ROCK2, involving residues Ala102, Phe103, Leu123, and Phe136. These interactions are consistent with the co-crystal ligand (Appendix A) and other most active compounds within the training set, all of which have pIC_50_ values above 8.00 (e.g., ChEMBL1922128, ChEMBL1922034, ChEMBL1922042, ChEMBL1922125, RKI1447, ChEMBL1922043). Notably, hydrogen bonds with Phe103 are absent in the least active molecules (e.g., ChEMBL2023148—Figure 5B, ChEMBL2023161—Appendix A, ChEMBL2023159—Appendix A), likely due to the absence of hydrogen-donating substituents on the phenyl ring. Collectively, these findings suggest that meta-substitution of the phenyl ring may be critical for ROCK2 inhibitory activity, contributing through both hydrophobic interactions with the P-loop and hydrogen bond formation.

Finally, a hydrogen bond was found between the oxygen atom of the carbonyl group in the linker of the most active compounds, ChEMBL1922128 and Lys121 (Figure 5A). Similar hydrogen bonds are also observed in other highly active compounds, e.g., RKI1447 (Appendix A), where the nitrogen atom of the ureido group forms a hydrogen bond with Lys121, as well as between the co-crystal ligand VFA501 and ROCK2 (Appendix A). These results emphasize the importance of hydrogen-donating groups in the linker region of the molecule for effective ROCK2 inhibition.

#### 3.1.3. Design and Synthesis of Novel Potential ROCK Inhibitors

Based on the findings from 3D-QSAR (ROCK1) and 3D-QSAR (ROCK2), as well as molecular docking studies, we can conclude that the inhibition of ROCK1 and ROCK2 is significantly influenced by the presence of a hinge-binding moiety—a heterocyclic structure containing a hydrogen-accepting group. The critical role of this molecular feature is evidenced by var323 (ROCK1), var328 (ROCK1), var254 (ROCK2), and var459 (ROCK2). Furthermore, molecular docking studies have demonstrated that the formation of hydrogen bonds between the hinge-binding moiety and key amino acid residues—specifically Glu154 and Met156 in ROCK1 and Glu170 and Met172 in ROCK2—is essential for the inhibition of these enzymes. Moreover, introducing a linker containing hydrogen bond-donating or accepting groups appears to enhance the overall inhibitory activity against ROCK1 and ROCK2. This is supported by the positive influence of var459 (ROCK2) and the hydrogen bonds formed with Lys105 and Asp232 in ROCK1, as well as Lys121 in ROCK2. The introduction of a phenyl ring attached to the linker of the scaffold can enhance the inhibitory effect of ROCK1/ROCK2 inhibitors through hydrophobic interactions with key residues such as Ala103, Val137, Leu205, Met153, Phe120, Val90, and Gly85 in ROCK1 and the P-loop in ROCK2. This enhancement is also supported by the 3D-QSAR model for ROCK1, which shows that an optimal distance between the hydrogen-accepting group of the heterocycle and the phenyl ring has a positive effect on the activity of ROCK1 inhibitors, as observed by var323 and var328. Similarly, the positive influence of this structural feature is confirmed by var355 in the 3D-QSAR model for ROCK2.

Moreover, the incorporation of hydrogen-accepting groups as substituents on the phenyl ring can increase the activity of ROCK1 and ROCK2 inhibitors by facilitating hydrogen bonding with Phe87 in ROCK1 and Phe103 in ROCK2. This positive correlation between meta- and para-substitution on the phenyl ring and the inhibitory activity of ROCK1/ROCK2 inhibitors is confirmed by 3D-QSAR models, in particular by the positive influence of var573, var383, and var164 in ROCK1 and var597 and var477 in ROCK2 on the overall inhibitory activity.

Based on the results of our 3D-QSAR and molecular docking studies for ROCK1 and ROCK2, we designed a series of novel ROCK inhibitors, as shown in Figure 6. The hinge-binding moieties selected for these compounds include isoquinoline, pyridine, and pyridine-4-phenyl groups, all of which have heterocyclic structures with hydrogen-accepting nitrogen atoms consistent with our computational predictions. These moieties are linked to a scaffold containing both hydrogen-donating (nitrogen of amide) and hydrogen-accepting (carbonyl) groups, consistent with the conclusions identified in our 3D-QSAR models and docking analyses for ROCK1 and ROCK2. Furthermore, the incorporation of a phenyl ring into the linker region increases the potency of the inhibitors by promoting hydrophobic interactions with key residues in ROCK1 (Ala103, Val137, Leu205, Met153, Phe120, Val90, and Gly85) and ROCK2 (P-loop). Our studies have also shown that substituents with hydrogen-accepting properties on the phenyl ring further enhance the inhibitory potential of these compounds. To exploit this, we introduced fluoro, hydroxy, and methoxy groups at different meta- and para-positions of the phenyl ring. In this way, nine new potential ROCK inhibitors were developed (Figure 6).

The binding modes and key interactions of the designed compounds were evaluated by molecular docking studies on both ROCK isoforms. Figure 7 illustrates the binding mode and key interactions of one of these designed molecules, **C-19**. All designed compounds formed a hydrogen bond between the nitrogen atom of the hinge-binding moiety (heterocycle) and the Met156 amino-acid residue in ROCK1 or the Met172 in ROCK2, which are crucial for ROCK1/ROCK2 inhibition.

The designed ROCK inhibitors were synthesized by the routes shown in Figure 8. The homopiperazine derivatives were prepared by the reaction of isoquinoline-5-sulfonic acid, 3-(4-pyridyl) acrylic acid, or 4-(4-pyridyl) benzoic acid with homopiperazine. The benzyl bromide derivatives were synthesized from the corresponding benzaldehydes in a two-step process: reduction to the alcohol with NaBH_4_, followed by nucleophilic substitution with PBr_3_. The final compounds, potential ROCK inhibitors, were obtained by a nucleophilic substitution reaction between the homopiperazine and benzyl bromide derivatives, obtaining the products in moderate yield. The final compounds were purified by flash chromatography (DCM then DCM = 9:1), and their structures were confirmed by NMR and mass spectrometry (Appendix B and Appendix A).

### 3.2. Enzyme Assays

The synthesized ROCK inhibitors were screened for their activity against ROCK1 and ROCK2 using a commercial in vitro biochemical assay kit. Initially, a preliminary screening was performed to evaluate the inhibitory effects of the synthesized compounds on ROCK2, with assays conducted at a concentration of 10 µM (Figure 9A). Only two compounds, **C-19** and **C-22**, exhibited greater than 50% inhibition of ROCK2, 52.03% and 72.45%, respectively. Consequently, the inhibitory effects of these two compounds on ROCK1 were assessed. Compound **C-19** demonstrated significant inhibition of ROCK1 at 72.64%, whereas **C-22** showed less than 50% inhibition, with an activity of 27.36% (Figure 9A).

The IC_50_ value for compound **C-19** was subsequently determined, revealing moderate inhibition of both ROCK1 and ROCK2, with IC_50_ values of 4.22 µM and 6.13 µM, respectively (Figure 9B). Due to its promising inhibitory profile (Table 3), compound **C-19** was selected for further detailed biological evaluation.

### 3.3. Biological Evaluation

#### 3.3.1. Cytotoxic Effect of Newly Designed and Synthesized ROCK Inhibitors

This study examined the cytotoxic effect of newly developed and synthesized ROCK inhibitors (**C-19** and **C-22**) against four cancer cell lines (MDA-MB-231, HCC1937, Mia Paca-2, and Panc-1). A selective ROCK inhibitor, fasudil, was used as a positive control in all in vitro assays. After 72 h of treatment with **C-19**, a dose-dependent cytotoxic effect was obtained against both pancreatic cancer cell lines, with no effect on breast cancer cell lines. The IC_50_ values obtained were 19.10 μM and 35.03 μM for Mia PaCa-2 and Panc-1, respectively, indicating a higher sensitivity of Mia PaCa-2 cells. Meanwhile, **C-22** and fasudil treatment showed no cytotoxic effect on the tested cancer cells (Table 4 and Figure 10).

The toxicity of **C-19** was then investigated in a normal, i.e., non-transformed cell line (MRC-5), in order to determine the selectivity index (Si) of the molecules. Compound **C-19** showed excellent selectivity towards MiaPaCa-2 with a Si of 4.1 and moderately good selectivity towards Panc-1 (2.2) (Table 4).

#### 3.3.2. Newly Designed and Synthesized ROCK Inhibitor **C-19** Induces Early Apoptosis in Mia PaCa-2 and Panc-1 Cells

Exposure to the newly synthesized ROCK inhibitor **C-19** at a concentration of 20 μM induced early apoptosis in a population of over 20% of Mia-PaCa-2 (Figure 11A,B) and approximately 15% of Panc-1 cells (Figure 11C,D) after 48 h of treatment, as determined by flow cytometry using Annexin V-FITC/PI staining. The percentage of apoptotic cells after treatment with the tested compound is significantly higher in both pancreatic cell lines compared to the untreated control cells (*p* ≤ 0.05 for Panc-1, *p* ≤ 0.01 for Mia PaCa-2) and compared to the control—fasudil.

#### 3.3.3. Newly Designed and Synthesized ROCK Inhibitor **C-19** Induces Changes in Cell Cycle Phase Distribution in Mia PaCa-2 and Panc-1 Cells

To evaluate the potential antiproliferative effect of the synthesized **C-19** inhibitor, we performed flow cytometry analysis to assess changes in cell cycle distribution at IC_50_ concentrations. Treatment of Mia PaCa-2 and Panc-1 cells with ROCK inhibitors (**C-19** and fasudil) at a concentration of 20 μM for 48 h resulted in slight changes in cell cycle phase distribution (Figure 12). Exposure to **C-19** for 48 h slightly increased the percentage of Mia-PaCa-2 cells in the sub-G1 phase, followed by a decrease in cells in the G0/G1 phase. The same changes in the distribution of cell cycle phases were caused by exposure of Mia PaCa-2 cells to fasudil (Figure 12A,B). Treatment of Panc-1 cells with **C-19** slightly increased the percentage of cells in the G0/G1 phase, while treatment of Panc-1 cells with fasudil did not cause significant changes in the cell cycle (Figure 12C,D).

#### 3.3.4. Newly Designed and Synthesized ROCK Inhibitor **C-19** Induces a Decrease of Mitochondrial Membrane Potential in Mia Paca-2 Cells

Since activation of the intrinsic apoptotic pathway is characterized by alterations in mitochondrial membrane potential, the effects of **C-19** and fasudil on this potential were investigated. Treatment with the newly synthesized ROCK inhibitor **C-19** at a concentration of 20 μM resulted in a significant decrease in mitochondrial membrane potential in Mia PaCa-2 (*p* ≤ 0.01) (Figure 13A). In contrast, no change in mitochondrial membrane potential was observed in Panc-1 cells after 24 h of treatment compared to control (Figure 13B). Exposure to fasudil at a concentration of 20 μM for 24 h resulted in a slight polarization of mitochondria with higher membrane potential in both cell lines (Figure 13A,B).

#### 3.3.5. Newly Designed and Synthesized ROCK Inhibitor **C-19** Induces an Increase of Total ROS in Panc-1 Cells

Cancer cells are characterized by an excessive production of reactive oxygen species (ROS), which are predominantly found in the mitochondria. These ROSs contribute to tumor progression by affecting gene expression, causing genomic instability, and modulating various signaling pathways [56]. Consequently, the effects of **C-19** and fasudil on both total and mitochondrial ROS levels are being evaluated to understand their impact on these processes better. After 24 h treatment with **C-19** (concentration of 20 μM), a significant increase in the total ROS level was observed in Panc-1 cells (*p* ≤ 0.001) compared to control (Figure 14B). Meanwhile, 24 h treatment with fasudil (concentration of 20 μM) caused a significant increase in the total ROS level in Mia Paca-2 cells compared to the control (*p* ≤ 0.001) (Figure 14A). A 24 h treatment with the tested ROCK inhibitors did not affect the change in mitochondrial ROS levels in either pancreatic cell line.

#### 3.3.6. Analyses of the Efficacy of **C-19** in a 3D Cancer Model

It is now recognized that three-dimensional (3D) cancer models more accurately mimic the tissue environment and cell behavior observed in vivo compared to two-dimensional (2D) models. This advance has improved the precision of drug testing and significantly reduced the failure rate of new drugs in clinical trials [57].

The size of MiaPaCa-2 spheres treated with the tested **C-19** as well as fasudil decreased significantly compared to the untreated control (Figure 15A,B), while only fasudil treatment at the concentration used in the 2D model (20 μM) significantly decreased the viability of MiaPaCa-2 spheres. The size of Panc-1 spheres treated with **C-19** did not significantly decrease compared to the untreated control (Figure 15A,B) but significantly reduced the viability of Panc-1 spheres at the concentration used in the 2D model (20 μM). Fasudil treatment of Panc-1 spheres did not affect size and viability at a concentration of 20 μM (Figure 15A,B).

In another set of experiments, the ability of the tested compounds to inhibit tumorsphere formation was investigated. MiaPaCa-2 and Panc-1 cells were cultured with the compounds (at a concentration of 20 μM) under sphere culture conditions for 4 days (Figure 15C,D). After 4 days, the untreated cells formed spheres with a densely packed morphology (Figure 15C,D). Incubation of MiaPaCa-2 cells with **C-19** and fasudil at the concentration used in the 2D model (20 μM) for 4 days under sphere culture conditions significantly inhibited both sphere formation and viability (Figure 15C,D). In contrast, compound **C-19** at a concentration of 20 μM caused a strong cytotoxic effect on Panc-1 tumorspheres as well as a significant inhibition of Panc-1 sphere formation (Figure 15C,D). Fasudil showed none of the examined effects on treated Panc-1 spheres (Figure 15C,D).

#### 3.3.7. Synthetized ROCK Inhibitors Inhibit Cell Migration and Invasion In Vitro

Cancer cells must acquire cell motility capabilities and a substantial cytoskeleton reorganization to invade neighboring and distant tissues and organs. They interact with the external microenvironment to guide cell movements and form cellular structures, such as lamellipodia and filopodia, needed for cell migration and invasion [2]. In this sense, ROCK proteins are key cytoskeleton regulators and affect cellular functions such as cell proliferation and motility. Also, ROCK expression is elevated in several cancer types and is associated with tumor progression and metastasis [58]. Moreover, targeting ROCK seems to be a promising strategy to reduce cancer cell migration and invasion, thus inhibiting the metastasis process [12].

Chemically similar isoquinoline-based ROCK inhibitors **C-19** and **C-22** were analyzed to determine their anti-migratory and anti-invasive activities on triple-negative HCC1937 and MDA-MB-231 breast cancer cells and pancreatic ductal adenocarcinoma Panc-1 and MiaPaca-2 cell lines. First, a wound-healing assay with the epithelial-like HCC1937 and Panc-1 cells was performed. This methodology determines directional cell migration in vitro [59]. All the drug treatments were performed according to the IC_50_ values from Table 4 and did not significantly affect cell viability during the 24 h testing period As shown in Figure 16A,B, **C-19** and **C-22** ROCK inhibitors provoke a reduced wound cell closure comparable to fasudil. Next, we determined ROCK inhibitors **C-19** and **C-22** on the mesenchymal-like MDA-MD-231 and the pleomorphic- and spindle-shaped MiaPaca-2 cell lines for transwell migration and invasion in vitro. Similarly to results obtained in the wound healing assay, both **C-19** and **C-22** exert anti-invasive (Figure 17A,C) and anti-migratory activities (Figure 17B,D) comparable to fasudil, as is demonstrated by a decrease in the cancer cell capacity to cross transwell membranes to the bottom of membranes in both the invasion and migration assays. Therefore, these results suggest that **C-19** and **C-22**, likely inhibiting ROC1 and ROCK2, can effectively regulate tumoral cell migration and invasion, making them promising agents for targeting metastatic tumors.

#### 3.3.8. Future Perspectives

Compound **C-19** exhibited the most potent anti-cancer properties among the newly developed compounds, with IC_50_ values of 19.10 μM and 35.03 μM against Mia PaCa-2 and Panc-1 cell lines, respectively. In contrast, compound **C-22** showed no significant activity against the tested cancer cell lines. Treatment with **C-19** significantly increased the percentage of apoptotic cells in both pancreatic cancer cell lines. In addition, **C-19** treatment affected the cell cycle by notably increasing the percentage of Mia-PaCa-2 cells in the sub-G1 phase and correspondingly reducing the fraction of cells in the G0/G1 phase. In Panc-1 cells, a slight increase in the proportion of cells in the G0/G1 phase was observed after treatment with **C-19**. Moreover, **C-19** and **C-22**, the compounds with the strongest ROCK inhibitory effects among the synthesized compounds, showed potent anti-migratory and anti-invasive effects comparable to the well-known ROCK inhibitor fasudil. These results suggest that these new compounds hold promise as potential therapeutics for treating metastatic tumors. Due to the high degree of similarity of the kinase structures, it is likely that the compounds we synthesized also have some activity on other kinases, which should be investigated in further studies.

In line with one of the main objectives of this study, we propose a novel multi-target approach that utilizes the structural framework of the recently developed ROCK inhibitors **C-19** and **C-22**. Building on the findings of Djokovic N. et al. (2023) [21], who highlighted the synergistic effects between ROCK inhibitors and histone deacetylase (HDAC) inhibitors and considering the potent anti-migratory and anti-invasive properties of **C-19** and **C-22**, the development of dual HDAC/ROCK inhibitors represents a promising avenue to improve antimetastatic efficacy.

The design of these multi-target HDAC/ROCK inhibitors was based on a pharmacophore fusion strategy in which the pharmacophore of the ROCK inhibitors developed in this study was connected to a zinc-binding group by different linkers, as shown in Figure 18.

A molecular docking study was performed to investigate the binding modes and key interactions between the designed molecules and ROCK1, ROCK2, and HDAC6. Two compounds, **D-1** and **D-7**, were found to establish all critical interactions required for the inhibition of all three enzymes. The binding modes of these compounds are shown in Figure 19.

Both designed molecules formed complexes with the zinc ion within catalytic domain 2 of HDAC6, which were further stabilized by hydrogen bonding with His610 (**D-1**) and His610 and His611 (**D-7**). This interaction pattern is considered crucial for the effective inhibition of HDAC6. In addition, van der Waals interactions were observed with Phe680, Phe620, and Phe679, which may also contribute to the inhibition of HDAC6 (Figure 19).

**D-1** and **D-7** also formed key hydrogen bonds with ROCK1 (Met156) and ROCK2 (Met172) alongside hydrophobic interactions with the P-loop of ROCK2 and various residues in ROCK1, including Ala103, Leu205, Lys105, and Gly85 (Figure 19). These interactions may likely correlate with the inhibitory effect of the newly developed molecules against ROCK. Furthermore, hydrogen bonds formed by the carbonyl group of the linker with Phe87 (ROCK1) and Phe103 (ROCK2), which are known to be important for the inhibition of these enzymes, were identified (Figure 19). Although further studies are needed to experimentally validate our results from molecular docking, the findings to date suggest that this multitarget strategy has significant potential and could represent an important milestone in the development of future cancer treatments.

## 4. Conclusions

This study has significantly expanded our understanding of ROCK inhibitors and their potential as therapeutic agents in cancer treatment. Through extensive computational and experimental approaches, we have identified key structural features critical for the inhibition of ROCK1 and ROCK2. Our results emphasize the importance of hydrogen bonds, hydrophobic interactions and the introduction of specific linker and phenyl ring modifications in enhancing the inhibitory effect of ROCK inhibitors.

Among the synthesized compounds, **C-19** showed the most potent anti-cancer activity by effectively inducing apoptosis and cell cycle arrest in pancreatic cancer cell lines while exhibiting strong anti-migratory and anti-invasive activity. These promising results suggest that **C-19** and, to a lesser extent, **C-22** could serve as valuable leads for developing new therapeutic agents against metastatic tumors.

Building on the synergistic potential of ROCK and HDAC inhibitors, we also propose a novel multi-target approach that utilizes the structural scaffold of **C-19** and **C-22** to develop dual HDAC/ROCK inhibitors. This approach, which combines the ROCK and HDAC pharmacophores, can potentially increase antimetastatic efficacy and overcome the challenges of chemoresistance commonly encountered in single-agent cancer therapy. In summary, the results of this study provide a solid framework for the future development of multi-target ROCK inhibitors, which may represent a promising new strategy in the fight against cancer.

## Figures and Tables

**Figure 1 pharmaceutics-16-01250-f001:**
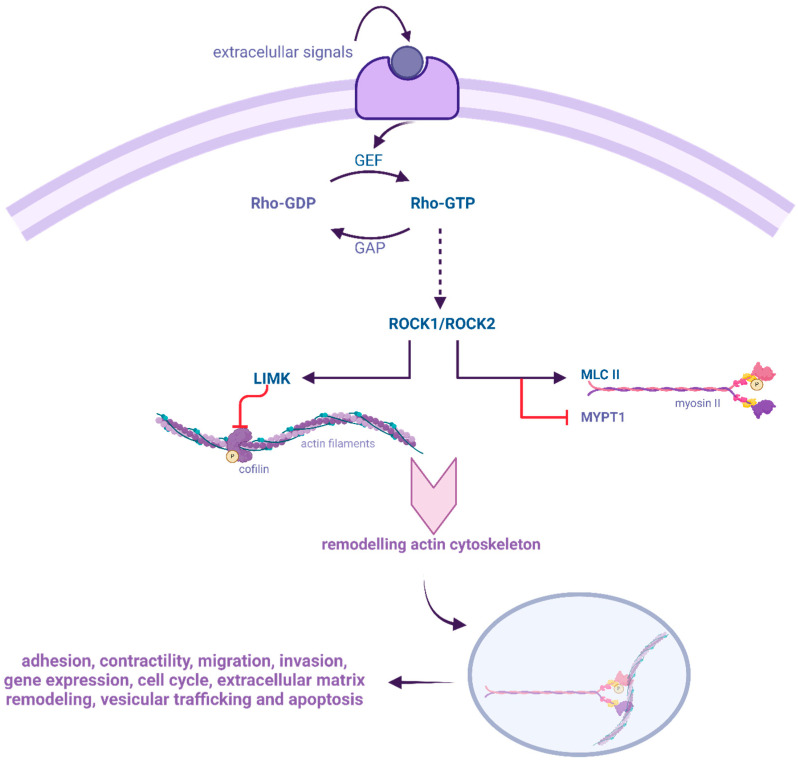
Rho-GTPase-ROCK-downstream effectors signaling pathway. Created with BioRender.com.

**Figure 2 pharmaceutics-16-01250-f002:**
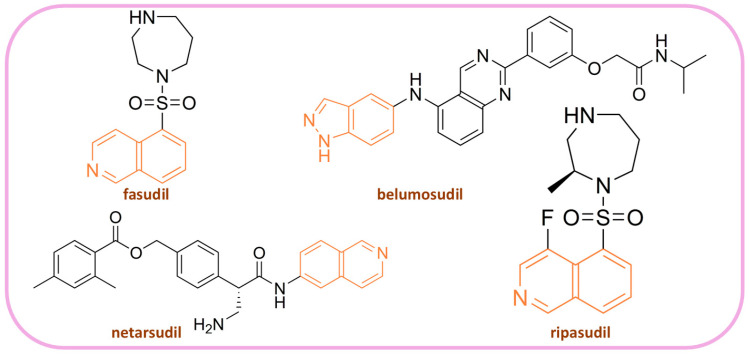
Structures of already FDA-approved ROCK inhibitors, hinge binding domain is labeled orange.

**Figure 3 pharmaceutics-16-01250-f003:**
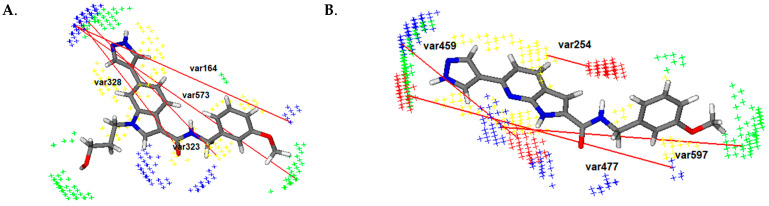
(**A**) Favorable Var164: N1-N1, favorable Var323: DRY-N1, favorable Var328: DRY-N1 and favorable Var573: N1-TIP of compound ChEMBL1922128 (3D-QSAR (ROCK1) model). (**B**) Favorable Var254: DRY-O, favorable Var459: O-N1, favorable Var477: O-N1 and favorable Var597: N1-TIP of compound ChEMBL1921034 (3D-QSAR (ROCK2) model); the hydrophobic regions (DRY) are labeled in yellow, H-bond donor regions in red, H-bond acceptor regions in blue and the steric hot spots (TIP) are presented in green.

**Figure 4 pharmaceutics-16-01250-f004:**
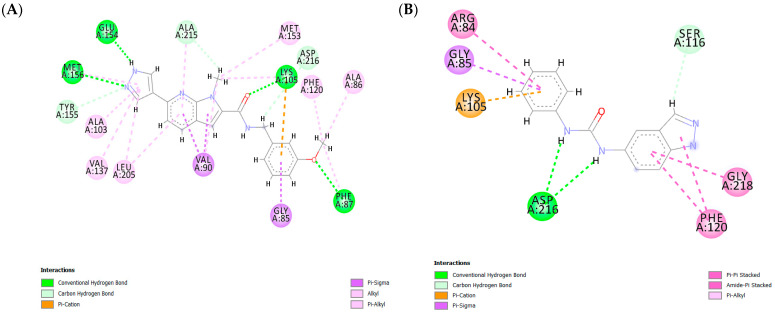
(**A**) Diagram showing the key interactions between CHEMBL1922035 and ROCK1 (PDB: 6E9W) and (**B**) between CHEMBL2023148 and ROCK1 (PDB: 6E9W). The ChemPLP score for CHEMBL1922035 is 83.36, while the ChemPLP score for CHEMBL2023148 is 60.97.

**Figure 5 pharmaceutics-16-01250-f005:**
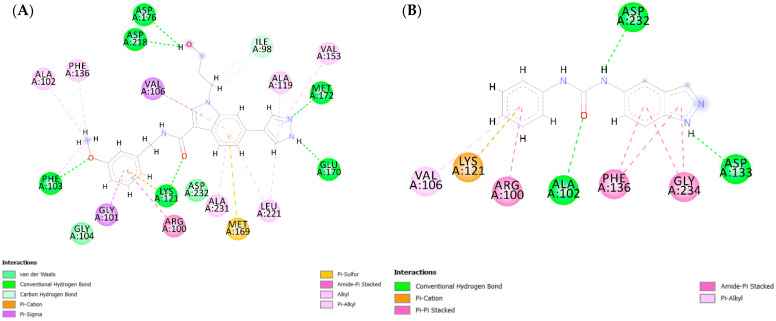
(**A**) Diagram showing the key interactions between CHEMBL1922128 and ROCK2 (PDB: 7JNT) and (**B**) between CHEMBL2023148 and ROCK2 (PDB: 7JNT). The ChemPLP score for CHEMBL1922128 is 93.01, while the ChemPLP score for CHEMBL2023148 is 50.26.

**Figure 6 pharmaceutics-16-01250-f006:**
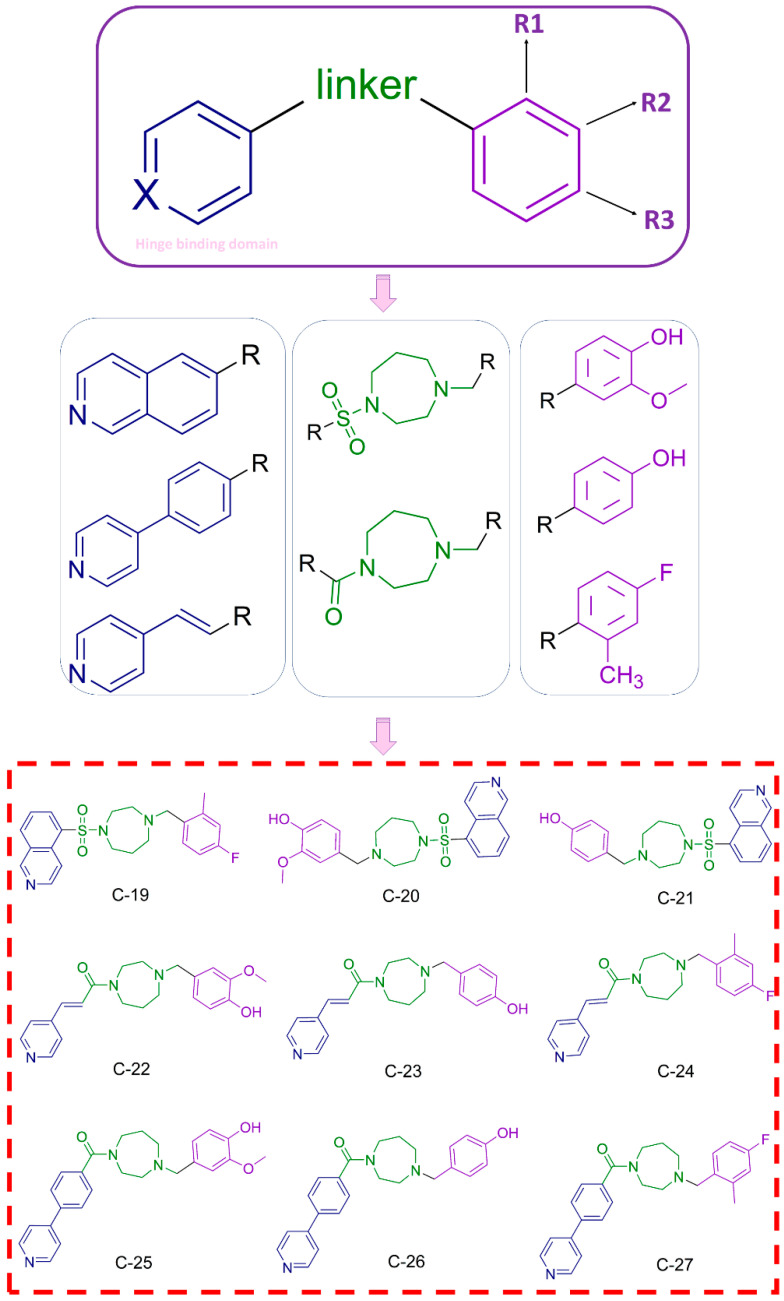
Design of novel ROCK inhibitors, the hinge binding domain is marked in blue, the linker in green, and the substituted phenyl ring in purple.

**Figure 7 pharmaceutics-16-01250-f007:**
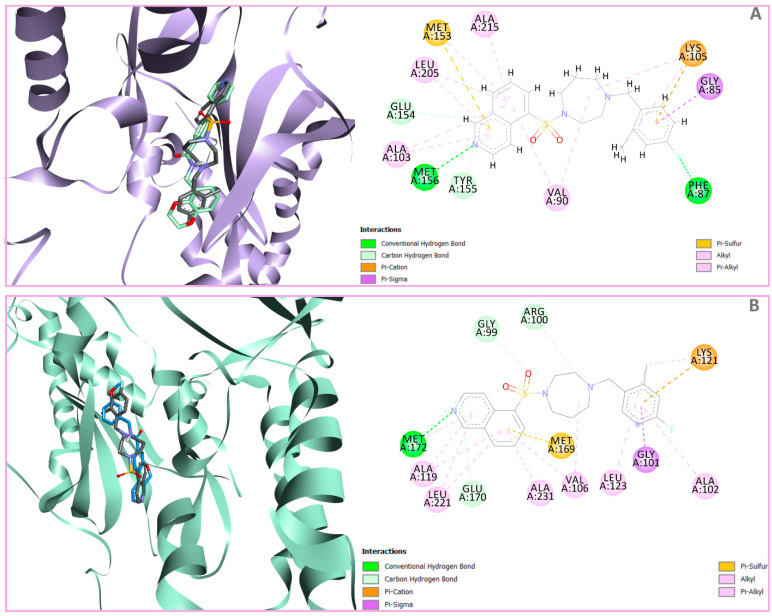
(**A**) The binding position of the novel ROCK inhibitor **C-19** (grey) with a ChemPLP score of 80.32 and the co-crystal ligand J0P501 (green) with a ChemPLP score of 82.68 in the active site of ROCK1 (left), together with a 2D representation of key **C-19**–ROCK1 interactions (right). (**B**) The binding position of **C-19** (grey) with a ChemPLP score of 82.82 and the co-crystal ligand VFA501 (blue) with a ChemPLP score of 82.46 in the active site of ROCK2 (left), accompanied by a two-dimensional (2D) diagram highlighting the key **C-19**–ROCK2 interactions. J0P501 and VFA501 are the co-crystal ligands of ROCK1 (PDB: 6E9W) and ROCK2 (PDB: 7JNT), respectively, providing a comparative analysis of their binding modes with those of the new ROCK inhibitor **C-19**.

**Figure 8 pharmaceutics-16-01250-f008:**
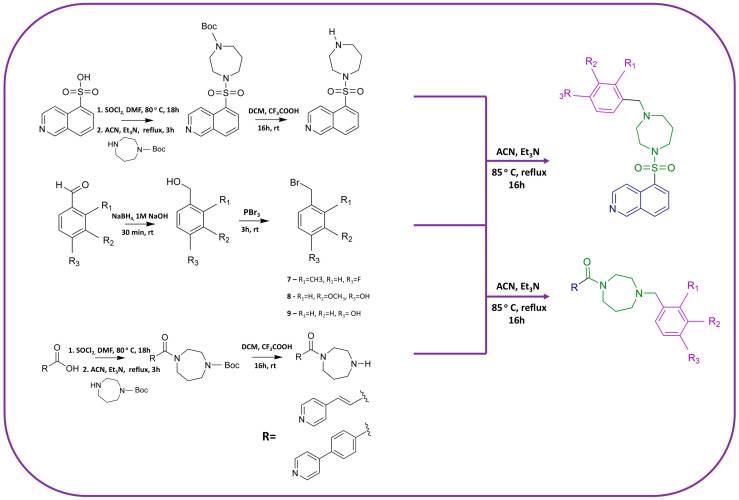
Synthetic routes for the preparation of the final compounds that are presented in Figure 6. The hinge binding domain is marked in blue, the linker in green, and the substituted phenyl ring in purple.

**Figure 9 pharmaceutics-16-01250-f009:**
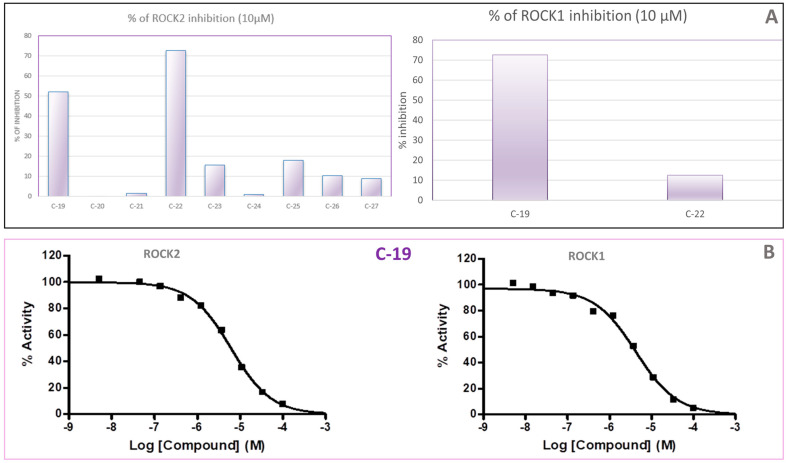
(**A**) Screening assays against ROCK2 and ROCK1 at 10 µM of each compound; (**B**) IC_50_ curves for **C-19** for ROCK1 and ROCK2.

**Figure 10 pharmaceutics-16-01250-f010:**
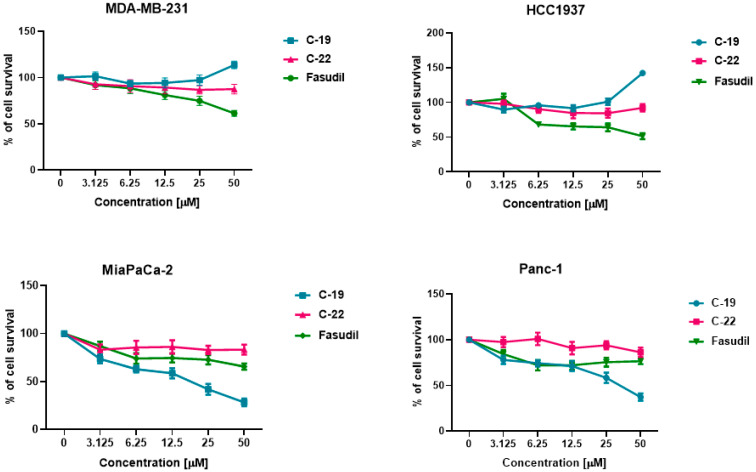
Effect of 72 h exposure to ROCK inhibitors on the viability of MDA-MB-231, HCC1937, Mia PaCa-2, and Panc-1 cell lines.

**Figure 11 pharmaceutics-16-01250-f011:**
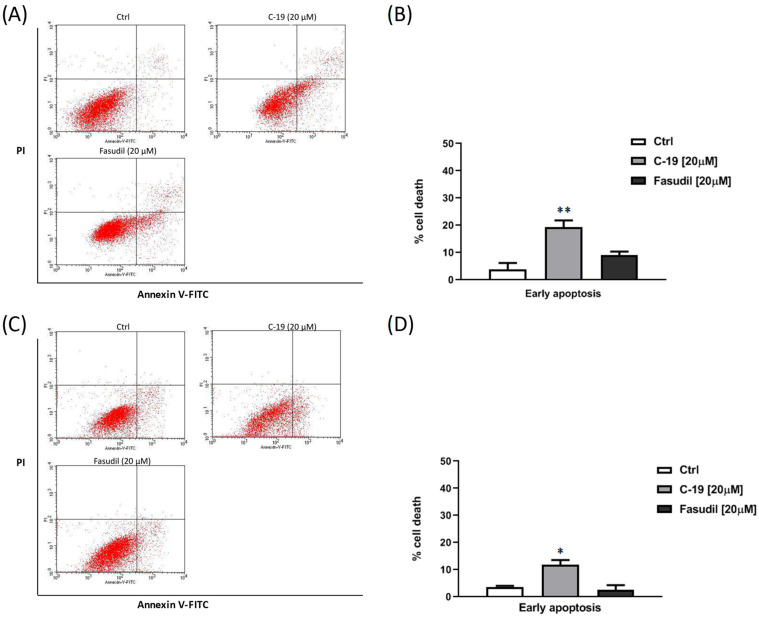
Effect of 48 h exposure to ROCK inhibitors (**C-19** and fasudil) on the induction of early apoptosis in Mia PaCa-2 (**A**,**B**) and Panc-1 (**C**,**D**) cells. Apoptosis was assessed by flow cytometry using the BD Pharmingen FITC Annexin V assay. A statistically significant difference between treatments was detected using the *t*-test: * *p* ≤ 0.05, ** *p* ≤ 0.01 (compared to the untreated control). The bar chart shows the results as mean ± SD of three independent experiments. Representative dot plots of Annexin V/PI-stained cells obtained by flow cytometry after 48 h treatment of MiaPaCa-2 (**A**) and Panc-1 (**C**) cells with the tested compounds.

**Figure 12 pharmaceutics-16-01250-f012:**
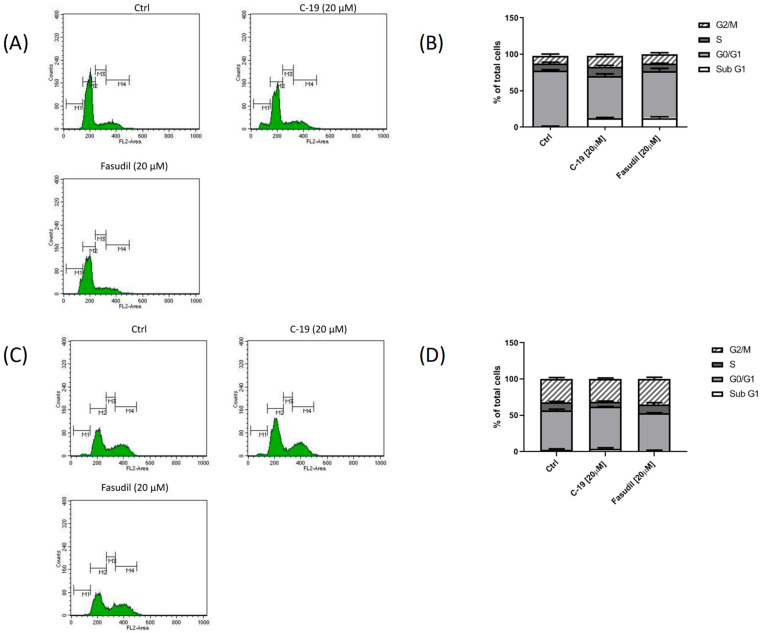
Effect of 48 h exposure to ROCK inhibitors (**C-19** and fasudil) on the changes in the cell cycle phase distribution of Mia PaCa-2 (**A**,**B**) and Panc-1 (**C**,**D**) cells. After 48 h of continuous exposure of pancreatic cancer cells to the tested ROCK inhibitors at a concentration of 20 μM, the cells were stained with propidium iodide and analyzed by flow cytometry. Representative histogram diagrams of cells stained with propidium iodide (PI) showing the cycle distribution after 48 h treatment of MiaPaCa-2 (**A**,**B**) and Panc-1 (**C**,**D**) cells with equimolar concentrations of **C-19** and fasudil. M1—apoptotic cells with DNA content corresponding to the sub-G1 fraction; M2—cells with DNA content corresponding to the G0/G1 phase; M3—cells with DNA content corresponding to the S phase; M4—cells with DNA content corresponding to the G2/M phase.

**Figure 13 pharmaceutics-16-01250-f013:**
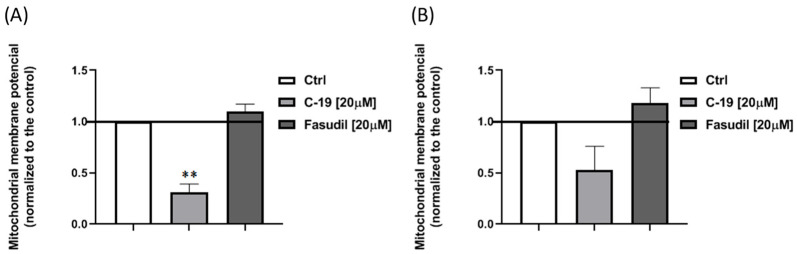
Effect of 24 h exposure to ROCK inhibitors (**C-19** and fasudil) on the changes in mitochondrial membrane potential of Mia PaCa-2 (**A**) and Panc-1 (**B**) cells. After 24 h of treatment with the tested ROCK inhibitors at a concentration of 20 μM, pancreatic cancer cells were stained with JC-1, and the change in mitochondrial membrane potential was analyzed by flow cytometry. A statistically significant difference between treatments was determined by *t*-test: ** *p* ≤ 0.01 (compared to the untreated control).

**Figure 14 pharmaceutics-16-01250-f014:**
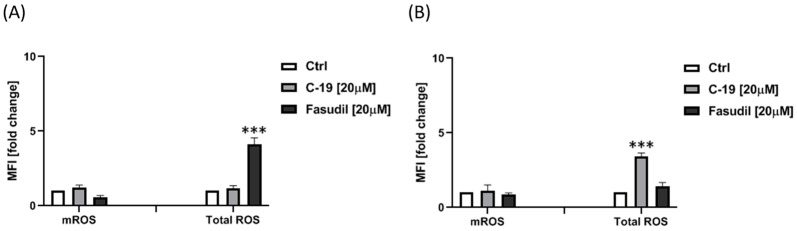
Effect of 24 h exposure to ROCK inhibitors (**C-19** and fasudil) on the levels of endogenous total and mitochondrial ROS in Mia PaCa-2 (**A**) and Panc-1 (**B**) cells. After 24 h of treatment with the tested ROCK inhibitors at a concentration of 20 μM, pancreatic cancer cells were stained with dichlorofluorescein and MitoSOX Red, and the changes in ROS concentration were analyzed by flow cytometry. A statistically significant difference between treatments was determined by a *t*-test: *** *p* ≤ 0.001 (compared to untreated control).

**Figure 15 pharmaceutics-16-01250-f015:**
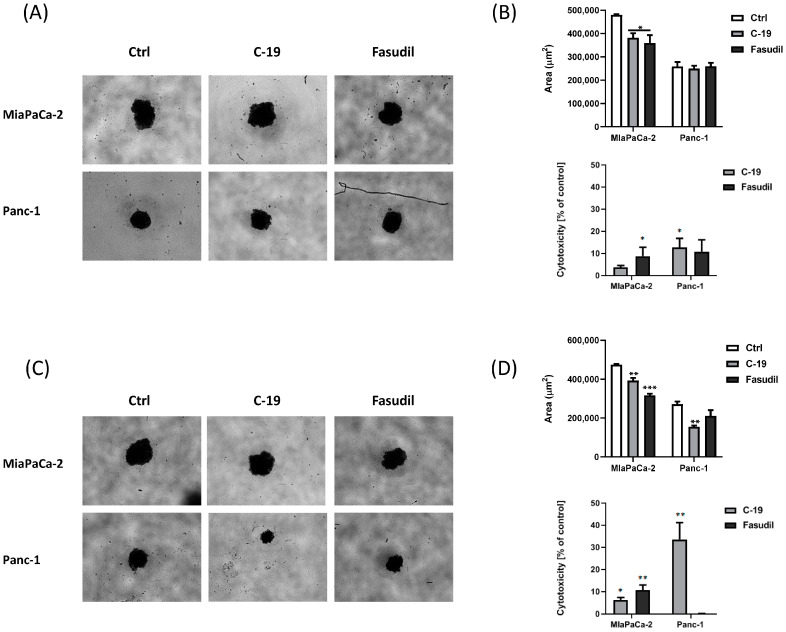
Inhibition of cell survival and growth of MiaPaca-2 and Panc-1 tumor spheres treated with synthesized compounds. After 4 d, formed spheres were treated with **C-19** compound and fasudil at equimolar concentration (20 µM) for 72 h. The formation and growth of the tumorsphere were examined and imaged with an Olympus CKX53, using a 4×/0.13 objective. Scale bar: 200 µm. (**A**) MiaPaca-2 and Panc-1 tumorspheres were observed under the bright field, and (**B**) sphere size was measured using ImageJ software, and the cytotoxicity of compounds toward the MiaPaCa-2 and Panc-1 tumor spheres was investigated by MTT assay. (**C**) MiaPaca-2 and Panc-1 cells were co-cultured with the **C-19** compound and fasudil (20 µM) in sphere culture conditions for 4 d. The formation and growth of tumor spheres were examined and imaged with an Olympus CKX53, using a 4x/0.4 objective. Scale bar: 200 µm. (**D**). A statistically significant difference between treatments was determined by a *t*-test: * *p* ≤ 0.05, ** *p* ≤ 0.01, *** *p* ≤ 0.001. The sphere size was measured using ImageJ software (ImageJ 1.54g) and cytotoxicity of compounds in the co-treatment with MiaPaCa-2 and Panc-1 tumorspheres was investigated by MTT assay.

**Figure 16 pharmaceutics-16-01250-f016:**
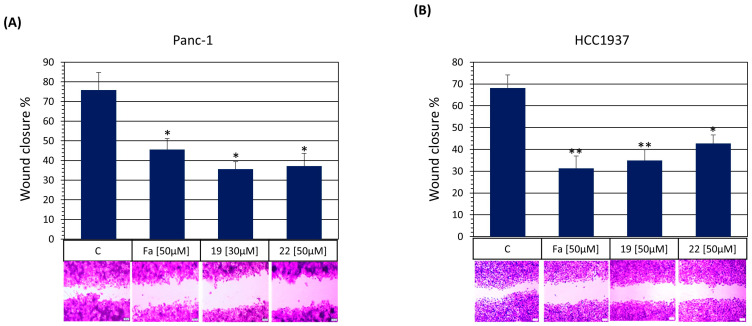
Anti-migratory of novel ROCK inhibitors (**C-19** and **C-22**) and fasudil on Panc1 cells (**A**) and HCC cells (**B**). A statistically significant difference between treatments was detected using the *t*-test: * *p* ≤ 0.05, ** *p* ≤ 0.01 (compared to the untreated control). The bar chart shows the results as mean ± SD of three independent experiments. Magnification 40×, Bar size: 50 µm.

**Figure 17 pharmaceutics-16-01250-f017:**
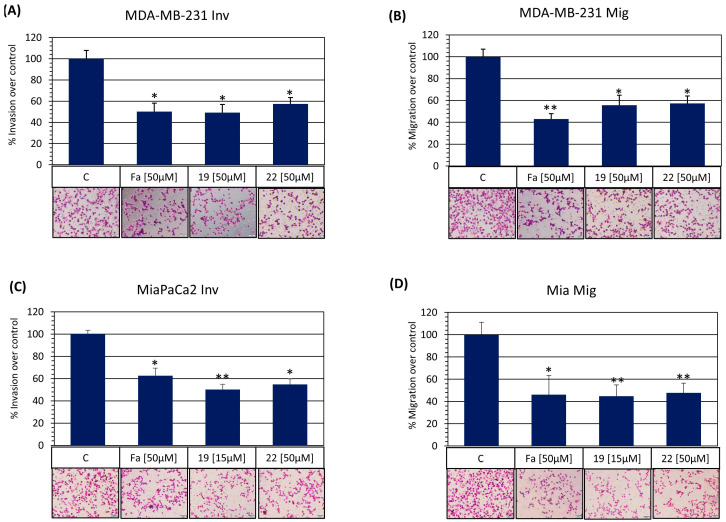
Anti-migratory and anti-invasive activities of novel ROCK inhibitors (**C-19** and **C-22**) and fasudil on MDA-MB-231 cells (**A**,**B**) and MiaPaca2 (**C**,**D**). A statistically significant difference between treatments was detected using the *t*-test: * *p* ≤ 0.05, ** *p* ≤ 0.01 (compared to the untreated control). The bar chart shows the results as mean ± SD of three independent experiments. Magnification 80×, Bar size: 100 µm. (Inv: invasion, Mig: Migration).

**Figure 18 pharmaceutics-16-01250-f018:**
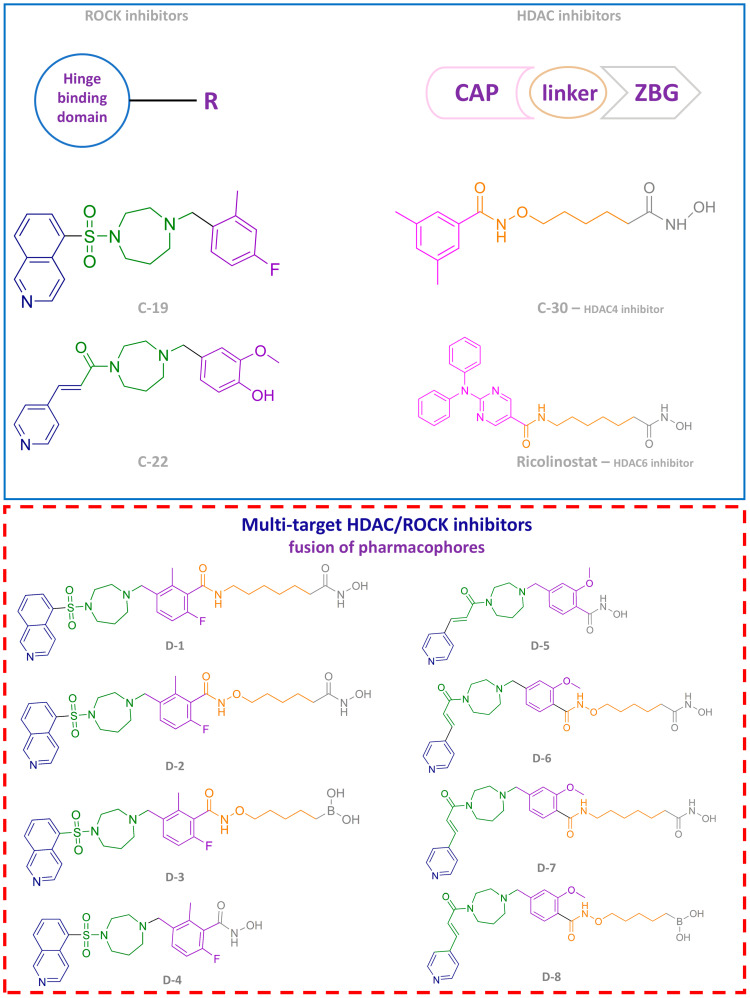
Schematic representation of the pharmacophore fusion strategy for the development of potential multi-target HDAC/ROCK inhibitors.

**Figure 19 pharmaceutics-16-01250-f019:**
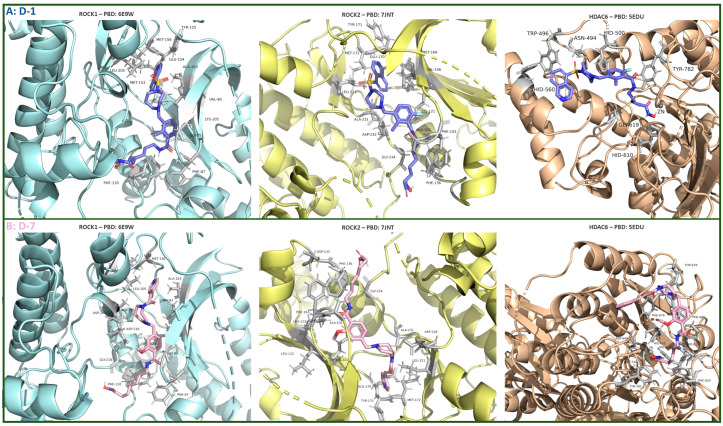
(**A**) Binding modes of the newly developed ROCK/HDAC multitarget inhibitor D-1 (blue) with ROCK1 (pale cyan), ROCK2 (pale yellow) and HDAC6 (wheat). Key interacting amino acid residues are highlighted in grey. (**B**) Binding modes of the newly developed ROCK/HDAC multitarget inhibitor D-7 (pink) with ROCK1 (pale cyan), ROCK2 (pale yellow) and HDAC6 (wheat). Key interacting amino acid residues are highlighted in grey.

**Table 1 pharmaceutics-16-01250-t001:** Results of developed 3D-QSAR (ROCK1) model.

**Internal Validation Parameters**				
	**R^2^**	**Q^2^_LOO_**	**RMSEE**	
	0.914	0.59	0.393	
Criteria	>0.9	>0.5		
**External Validation Parameters**				
	**R^2^_pred_**	**RMSEP**	**r_m_^2^**	**r^/2^_m_**
	0.792	0.561	0.759	0.573
Criteria	>0.6	≤2RMSEE	>0.5	>0.5
	$\bar{r_{m}^{2}}$	**Δr^2^_m_**	**(r^2^ − r^/2^_0_)/r^2^**	**k’**
	0.666	0.186	0.099	0.974
Criteria	>0.5	<0.2	<0.1	0.85 ≤ k’ ≤ 1.15

**Table 2 pharmaceutics-16-01250-t002:** Results of developed 3D-QSAR (ROCK2) model.

**Internal Validation Parameters**				
	**R^2^**	**Q^2^_LOO_**	**RMSEE**	
	0.933	0.54	0.327	
Criteria	>0.9	>0.5		
**External Validation Parameters**				
	**R^2^_pred_**	**RMSEP**	**r_m_^2^**	**r^/2^_m_**
	0.851	0.499	0.753	0.616
Criteria	>0.6	≤2RMSEE	>0.5	>0.5
	$\bar{r_{m}^{2}}$	**Δr^2^_m_**	**(r^2^ − r^/2^_0_)/r^2^**	**k’**
	0.684	0.137	0.032	0.963
Criteria	>0.5	<0.2	<0.1	0.85 ≤ k’ ≤ 1.15

**Table 3 pharmaceutics-16-01250-t003:** Predicted and experimental biological activity of **C-19**.

Cd	ROCK1			ROCK2		
	pIC_50_ pred	pIC_50_ exp	pIC_50_ exp − pIC_50_ pred	pIC_50_ pred	pIC_50_ exp	pIC_50_ exp − pIC_50_ pred
**C-19**	5.76	5.37	−0.39	5.61	5.21	−0.40

**Table 4 pharmaceutics-16-01250-t004:** Cytotoxic activities of tested compounds against MDA-MB-231, HCC1937, Mia PaCa-2, and Panc-1 cell lines.

Compound	IC_50_ (μM)				
	MDA-MB-231	HCC1937	MiaPaCa-2	Panc-1	MRC-5
**C-19**	>50 (n.d)	>50 (n.d)	19.10 ± 1.89 (4.1)	35.03 ± 4.43 (2.2)	78.25 ± 4.22
**C-22**	>50 (n.d)	>50 (n.d)	>50 (n.d)	>50 (n.d)	>50
Fasudil	>50 (n.d)	>50 (n.d)	>50 (n.d)	>50 (n.d)	92.02 ± 3.67

IC_50_ values (μM) were expressed as the mean value ± SD and represent results from three independent experiments n.d, not determined (IC_50_ > 50 μM).

## Data Availability

The raw data supporting the conclusions of this manuscript will be made available by the authors, without undue reservation, to any qualified researcher.

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
