# Peer review of "Development of Novel ROCK Inhibitors via 3D-QSAR and Molecular Docking Studies: A Framework for Multi-Target Drug Design"

_pharmaceutics, 2024, doi:10.3390/pharmaceutics16101250_

Round 1
Reviewer 1 Report
Comments and Suggestions for Authors
In the manuscript titled "Development of Novel ROCK Inhibitors via 3D-QSAR and Molecular Docking Studies: A Framework for Multi-Target Drug Design", the authors described the designs, synthesis, and evaluation of a series of ROCK1 and ROCK2 inhibitors. While the manuscript is well-written and the results thoroughly discussed, there is one major question that I would like the authors to answer/clarify: In the title, the authors mention "multi-target drug design", while briefly mentioning it in the intro, is listed under a few other potential combinations. Then, in section "3.3.8. Future perspectives", they mention the possibility of synergistic activity but the authors do not connect to the results in this manuscript. The connection is through a reference that previously described a synergistic effect for ROCK and HDAC inhibitors. The connection could have been done without all the experimental and computational data presented, as it is not based on any of the findings in this manuscript. None of the experimental results included HDAC.
Comments on the Quality of English LanguageThe manuscript is well-written and I have seen a few minor typos such as line 66 "LIM LIM kinases", and a few grammar issues such as lines 491-2, that is not clear and probably should be written stating that the increase ROCK inhibitor activity is due to the meta or para substituents and not the way that is presented.
Author Response
Dear reviewer,
Thank you for your comments which help us improve our manuscript. In the next sections you will find answers to your concerns:
- In the manuscript titled "Development of Novel ROCK Inhibitors via 3D-QSAR and Molecular Docking Studies: A Framework for Multi-Target Drug Design", the authors described the designs, synthesis, and evaluation of a series of ROCK1 and ROCK2 inhibitors. While the manuscript is well-written and the results thoroughly discussed, there is one major question that I would like the authors to answer/clarify: In the title, the authors mention "multi-target drug design", while briefly mentioning it in the intro, is listed under a few other potential combinations. Then, in section "3.3.8. Future perspectives", they mention the possibility of synergistic activity but the authors do not connect to the results in this manuscript. The connection is through a reference that previously described a synergistic effect for ROCK and HDAC inhibitors. The connection could have been done without all the experimental and computational data presented, as it is not based on any of the findings in this manuscript. None of the experimental results included HDAC.
Thank you for your valuable comment. In response, we have added a paragraph to the "Introduction" section of the manuscript providing a detailed explanation of the potential synergistic mechanisms between ROCK and HDAC inhibitors as evidenced by previous studies (see below). However, further studies are needed to fully understand the synergistic mechanisms between HDAC and ROCK inhibitors.
Lines 152-167: Some studies have shown that the signaling pathways of ROCK and histone deacetylase 6 (HDAC6) intersect during tumor progression, especially considering the cancer cell motility. ROCKs phosphorylate the Tubulin Polymerization Promoting Protein (TPPP1), which leads to inactivation of TPPP1 and impairs its interaction with HDAC6. As a result, HDAC6 is activated[32]. Therefore, inhibition of ROCK leads to increased tubulin acetylation, while actin cytoskeletal remodeling is also impaired - both essential processes for cancer cell movement. The dual inhibition of ROCK and HDAC6 can therefore achieve two effects: stabilization of the microtubule network by enhancing tubulin acetylation and prevention of cancer cell migration by disrupting actin cytoskeleton dynamics and stabilizing microtubule dynamics. In this way, one of the possible synergistic mechanisms between HDAC and ROCK inhibitors was revealed. However, further studies are needed to fully understand the synergistic mechanisms. All in all, we can assume that the integration of HDAC and ROCK inhibitory functions in a single molecule could potentially increase the efficacy of anticancer drugs by simultaneously targeting multiple signaling pathways involved in cancer cell proliferation, survival and metastasis, thus representing a new and effective strategy for cancer therapy.
The potential multitarget HDAC/ROCK inhibitors mentioned in the manuscript were developed based on the results of this study. In particular, the structures of the new ROCK inhibitors C-19 and C-22 served as the basis for a pharmacophore fusion strategy. In this approach, important structural features that are crucial for the inhibition of HDAC6 were integrated into the structures of the ROCK inhibitors (C-19 and C-22) developed in this study. To support this multitarget hypothesis, we have now performed molecular docking studies with the enzymes ROCK1, ROCK2 and HDAC6. Eight new compounds were developed (Figure 18), with D-1 and D-7 forming key binding interactions with all three enzymes (Figure 19, Section 3.3.8. Future perspectives). Although the present manuscript does not include experimental validation of HDAC inhibition, the next steps will involve synthesizing and evaluating these compounds to confirm the results of our molecular docking studies.
In line with the additional molecular docking study, we have added a few paragraphs to the Future Perspectives section.
Lines 981-985: A molecular docking study was performed to investigate the binding modes and key interactions between the designed molecules and ROCK1, ROCK2 and HDAC6. Two compounds, D-1 and D-7, were found to establish all critical interactions required for inhibition of all three enzymes. The binding modes of these compounds are shown in Figure 19.
Lines 992-1006: Both designed molecules formed complexes with the zinc ion within catalytic domain 2 of HDAC6, which were further stabilized by hydrogen bonding with His610 (D-1) and His610 and His611 (D-7). This interaction pattern is considered crucial for effective inhibition of HDAC6. In addition, van der Waals interactions were observed with Phe680, Phe620 and Phe679, which may also contribute to the inhibition of HDAC6 (Figure 19).
D-1 and D-7 also formed key hydrogen bonds with ROCK1 (Met156) and ROCK2 (Met172) alongside hydrophobic interactions with the P-loop of ROCK2 and various residues in ROCK1, including Ala103, Leu205, Lys105, and Gly85 (Figure 19). These interactions may likely correlate with the inhibitory effect of the newly developed molecules against ROCK. Furthermore, hydrogen bonds formed by the carbonyl group of the linker with Phe87 (ROCK1) and Phe103 (ROCK2), which are known to be important for the inhibition of these enzymes, were identified (Figure 19). Although further studies are needed to experimentally validate our results from molecular docking, the findings to date suggest that this multitarget strategy has significant potential and could represent an important milestone in the development of future cancer treatments
Figure 18. Schematic representation of the pharmacophore fusion strategy for the development of potential multi-target HDAC/ROCK inhibitors
Figure 19. (A) Binding modes of the newly developed ROCK/HDAC multitarget inhibitor D-1 (blue) with ROCK1 (pale cyan), ROCK2 (pale yellow) and HDAC6 (wheat). Key interacting amino acid residues are highlighted in grey. (B) Binding modes of the newly developed ROCK/HDAC multitarget inhibitor D-7 (pink) with ROCK1 (pale cyan), ROCK2 (pale yellow) and HDAC6 (wheat). Key interacting amino acid residues are highlighted in grey.
Comments on the Quality of English Language
The manuscript is well-written and I have seen a few minor typos such as line 66 "LIM LIM kinases", and a few grammar issues such as lines 491-2, that is not clear and probably should be written stating that the increase ROCK inhibitor activity is due to the meta or para substituents and not the way that is presented.
Thank you for drawing our attention to these errors. We have taken your comments into account and corrected the errors.
(Lines 502-503: Consequently, we can assume that the increase in ROCK1 inhibitor activity is due to the presence of meta- or para-substituents)

Reviewer 2 Report
Comments and Suggestions for Authors
It is an interesting work in design and biological evaluation, but there are issues with the presentation of certain results that diminish the quality of the work. Here are some comments to improve the manuscript.
Figures 3 and 5. They do not contribute anything. These graphs are incomprehensible. Please improve these images.
Figures 6 and 7. Add more information to the legend. Which of these compounds is the one crystallized in the respective PDB? The compound found in the crystal should be used as a reference to compare the pose, interactions, and scoring energy.
Figure 8. While the designed chemical structures are described here, I suggest including a table with the following information: a) Compound (C-19 - C-27), b) predicted biological activity, c) obtained experimental biological activity, and d) residual value (pIC50 exp - pIC50 pred).
Figure 9. Include the scoring function values for the reference and the designed compound in the legend.
Figure 10. There is no uniformity in the format of the molecules. Some chemical structures appear distorted. The reaction conditions are also not clearly displayed.
Author Response
Dear reviewer,
Thank you for your comments which help us improve our manuscript. In the next sections you will find answers to your concerns:
- Figures 3 and 5. They do not contribute anything. These graphs are incomprehensible. Please improve these images.
Thank you for your valuable comment. Figures 3 and 5 highlighted the variables that correlated most strongly with ROCK inhibitor activity. However, we agree that other figures may be more important for the manuscript. In response, we have moved Figures 3 and 5 to the Supplementary Information, where they can now be found (Section S1.1. 3D-QSAR study).
- Figures 6 and 7. Add more information to the legend. Which of these compounds is the one crystallized in the respective PDB? The compound found in the crystal should be used as a reference to compare the pose, interactions, and scoring energy.
Thank you for your insightful comment. We have added the figures depicting the binding modes of the co-crystal ligands (J0P501 – ROCK1 and VFA501 – ROCK2) to the Supplementary Information (Figures S1 and S7). We also revised Section 3.1.2 (Molecular docking study) to include a detailed description of the key interactions between the co-crystal ligands and ROCK1/ROCK2.
Lines 573-579: The nitrogen atom of the pyridine ring in the co-crystal ligand J0P501 (ChemPLP Score: 82.68, PDB: 6E9W) forms a key interaction with Met156, which contributes significantly to ROCK1 inhibition. In addition, the J0P501–ROCK1 complex is stabilized by hydrogen bonds, one between the carbonyl group of the ligand and Lys105 and another between the oxygen atom of 1,4-dioxane and Phe87. Numerous hydrophobic interactions with residues such as Leu205, Ala103, Val90, Ala215, Arg84, Met153 and Gly85 further facilitate the anchoring of J0P501 to the ROCK1 binding site (Figure S1).
Lines 618-625: The co-crystal ligand of ROCK2 - VFA501 (ChemPLP Score: 82.46, PDB:7JNT) formed a hydrogen bond with Met172, which is considered crucial for ROCK2 inhibition (Figure S7). In addition, two hydrogen bonds are formed between VFA501 and ROCK2, one between the carbonyl group of the linker and Lys121 and another between the methoxy group and Phe103. Finally, numerous Van der Waals interactions are described between VFA501 and various amino acid residues such as Val106, Val153, Met169, Ala119, Leu221, Arg100, Phe136, Ala102, etc., which positively correlate with ROCK2 inhibition (Figure S7).
We also compared these interactions with those of the other ROCK1 and ROCK2 inhibitors, as mentioned below:
Lines 596-601: These key interactions (both or one of them) are consistently observed between in co-crystal ligand J0P501 (Figure S1) as well as in other potent inhibitors across various clusters, such as RKI1447 (pIC50= 8.10, ChemPLP Score: 78.68, cluster II – Figure S2), ChEMBL1084892 (pIC50 = 7.85, ChemPLP Score: 68.26, cluster I – Figure S3), and ChEMBL2023160 (pIC50 = 5.85, ChemPLP Score: 64.36 cluster III – Figure S3).
Lines 611-614: In addition, the oxygen atom of the carbonyl linker and the methoxy group on the phenyl ring of ChEMBL1922035 form hydrogen bonds with Lys105 and Phe87, respectively, enhancing the anchoring of the compound in the enzyme (Figure 4-A), consistent with the co-crystal ligand J0P501.
Lines 621-625: Finally, numerous van der Waals and π-sigma interactions are observed between the pyrazole, 7-azaindole, isoquinoline or phenyl ring and various amino acid residues, including Ala103, Val137, Leu205, Met153, Phe120, Val90 and Gly85, which further stabilize the binding of compounds within the ROCK1 binding site. This interaction pattern is similar to that of the co-crystal ligand J0P501.
Lines 640-644: The most active compound in the ROCK2 training set, ChEMBL1922128, which belongs to cluster IV (pIC50= 9.00, ChemPLP Score: 93.01), forms two critical hydrogen bonds between the NH group of the pyrazole hinge-binding moiety and the amino acid residues Met172 and Glu170 that is consistent with the binding mode of co-crystal ligand VFA501 (Figures 5-A and S7).
Lines 657-663: The methoxy group at the meta-position of the phenyl ring (ChEMBL1922128-Figure 5-A) forms a hydrogen bond with Phe103, an interaction observed in other highly active compounds, such as ChEMBL1922125 (pIC50 = 8.30, ChemPLP Score: 90.84-Figure S11) and RKI1447 (pIC50= 8.21-Figure S9) as well as co-crystal ligand VFA501 (Figure S7). Additionally, the methoxy group of the phenyl ring engages in hydrophobic interactions with the P-loop of ROCK2, involving residues Ala102, Phe103, Leu123, and Phe136. These interactions are consistent with the co-crystal ligand (Figure S7).
Lines 674-676: Similar hydrogen bonds are also observed in other highly active compounds, e.g. RKI1447 (Figure S9), where the nitrogen atom of the ureido group forms a hydrogen bond with Lys121, as well as between the co-crystal ligand VFA501 and ROCK2 (Figure S7).
Furthermore, additional information has been included in the legends of Figures 6 and 7 (Figures 4 and 5 in the revised manuscript), as described below:
Figure 4: (A) Diagram showing the key interactions between CHEMBL1922035 and ROCK1 (PDB: 6E9W) and (B) between CHEMBL2023148 and ROCK1 (PDB: 6E9W). The ChemPLP score for CHEMBL1922035 is 83.36, while the ChemPLP score for CHEMBL2023148 is 60.97.
Figure 5: (A) Diagram showing the key interactions between CHEMBL1922128 and ROCK2 (PDB: 7JNT) and (B) between CHEMBL2023148 and ROCK2 (PDB: 7JNT). The ChemPLP score for CHEMBL1922128 is 93.01, while the ChemPLP score for CHEMBL2023148 is 50.26.
- Figure 8. While the designed chemical structures are described here, I suggest including a table with the following information: a) Compound (C-19 - C-27), b) predicted biological activity, c) obtained experimental biological activity, and d) residual value (pIC50 exp - pIC50 pred).
Thank you for your suggestion. The pIC50 value was determined specifically for compound C-19, as the percentages of inhibition of the other compounds were not above 50 for both ROCK1 and ROCK2. Therefore, we have included a table in the main manuscript showing both the predicted and experimentally determined biological activity for C-19 (Table 3). The predicted biological activities for the other compounds are listed in the Supplementary Information.
Table 3. Predicted and experimental biological activity of C-19
Cd |
ROCK1 |
ROCK2 |
||||
C-19 |
pIC50 pred |
pIC50 exp |
pIC50 exp - pIC50 pred |
pIC50 pred |
pIC50 exp |
pIC50 exp - pIC50 pred |
|
5.76 |
5.37 |
-0.39 |
5.61 |
5.21 |
-0.40 |
- Figure 9. Include the scoring function values for the reference and the designed compound in the legend.
Thank you for your comment. We have now included the scoring function values in the description of Figure 9 (Figure 7 in the revised manuscript), as outlined below:
Figure 9. A. The binding position of the novel ROCK inhibitor C-19 (grey) with a ChemPLP score of 80.32 and the co-crystal ligand J0P501 (green) with a ChemPLP score of 82.68 in the active site of ROCK1 (left), together with a 2D representation of key C-19–ROCK1 interactions (right). B. The binding position of C-19 (grey) with a ChemPLP score of 82.82 and the co-crystal ligand VFA501 (blue) with a ChemPLP score of 82.46 in the active site of ROCK2 (left), accompanied by a 2D diagram highlighting the key C-19–ROCK2 interactions. J0P501 and VFA501 are the co-crystal ligands of ROCK1 (PDB: 6E9W) and ROCK2 (PDB: 7JNT), respectively, providing a comparative analysis of their binding modes with those of the new ROCK inhibitor C-19.
- Figure 10. There is no uniformity in the format of the molecules. Some chemical structures appear distorted. The reaction conditions are also not clearly displayed.
Thank you for pointing out this error. We have now formatted all molecules with the ACS object settings. In addition, the reaction conditions have been corrected and are clearly presented in Figure 10 (Figure 8 in the revised manuscript).
Figure 8. Synthetic routes for the preparation of the final compounds that are presented in Figure 6. Hinge binding domain is marked in blue, linker in green and substituted phenyl ring in purple.

Reviewer 3 Report
Comments and Suggestions for Authors
This manuscript entitled “Development of Novel ROCK inhibitors via 3D-QSAR and Molecular Docking Studies: A Framework for Multi-Target Drug Design” conducted 3D-QSAR and Molecular Docking toward some ROCK inhibitors obtained from ChEMBL. As a result, authors found fundamental interactions of each part (hinge binding domain, linker, and substituted phenyl) of inhibitors and design a series of compounds (Figure 8). The following biological assays revealed that C-19 showed better biological profiles than that of fasudil, including anti-cancer activity, apoptosis induction, cell cycle arrest, and anti-migratory and anti-invasive activity. Authors also provided future perspectives toward multifunctionalized molecules against ROCK and HDAC. Considering the following points of view awaiting being addressed, this manuscript is recommended published in this journal.
1. Authors conducted 3D-QSAR and Molecular Docking to understand the crucial interactions and structural features of ROCK1 and ROCK2 inhibitors, but the designed molecules showing in Figures are like the derivatives of fasudil. It seems like that 3D-QSAR and Molecular Docking did not help authors find compounds with different chemotype and structures in Figure 8 are from modification of fasudil.
2. The enzymatic assays did not compare the activity of C-19 and C-22 with fasudil.
3. The concentration in Figure 12 should be “3.125, 6.25, 12.5”, instead of “3,125, 6,25, 12,5”.
4. Misspelling in Figure 2 (Ripasudil).
5. In “3.1.1. Results of 3D-QSAR study for ROCK1 and ROCK2”, authors cited a lot of codes from CHEMBL and describes the presence and absence of certain interactions. It is difficult to understand without structures. Please provide the structures of these compounds in the supporting information.
6. The description of Figure 9 should explain the roles of J0P501 and VFA501.
Comments on the Quality of English LanguageThe format of variable should be consistent (some variables are shown in bold, some are not).
Author Response
Dear reviewer,
Thank you for your comments which help us improve our manuscript. In the next sections you will find answers to your concerns:
- Authors conducted 3D-QSAR and Molecular Docking to understand the crucial interactions and structural features of ROCK1 and ROCK2 inhibitors, but the designed molecules showing in Figures are like the derivatives of fasudil. It seems like that 3D-QSAR and Molecular Docking did not help authors find compounds with different chemotype and structures in Figure 8 are from modification of fasudil.
Thank you for your thoughtful feedback. We understand your concerns regarding the similarity of the designed molecules to fasudil. However, based on the extensive 3D-QSAR and molecular docking studies for both ROCK1 and ROCK2, the design of novel ROCK inhibitors is firmly based on the results of these computational models.
Our studies have shown that inhibition of ROCK1 and ROCK2 is highly dependent on the presence of a heterocyclic hinge-binding moiety containing a hydrogen-accepting group that is critical for the formation of essential hydrogen bonds with key residues in the active sites. This is also confirmed by variables such as var323 and var328 for ROCK1 and var254 and var459 for ROCK2. Following the chemicals available in our laboratory, we selected isoquinoline, pyridine and pyridine-4-phenyl groups as hydrogen binding moieties.
In addition, we designed the novel inhibitors with linkers containing hydrogen bond donating and accepting groups (amide groups) to enhance the overall inhibitory activity, which is consistent with the conclusions from our QSAR and docking analyses. These parts of the linkers form hydrogen bonds with key residues such as Lys105 and Asp232 in ROCK1 and Lys121 in ROCK2. Furthermore, the introduction of a phenyl ring at the linker enhances the inhibitory effect through hydrophobic interactions with key residues in ROCK1 (Ala103, Val137, Leu205, Met153, Phe120, Val90 and Gly85) and in ROCK2 (the P-loop). The 3D-QSAR models supported these structural features and showed that an optimal distance between the heterocyclic group and the phenyl ring correlates with increased ROCK1 activity, as shown by var323, var328 and var355 for ROCK2.
Finally, the substituents on the phenyl ring (fluoro, hydroxy and methoxy groups) were carefully selected for their ability to enhance the inhibitory effect through hydrogen bonding with residues such as Phe87 in ROCK1 and Phe103 in ROCK2. The positions of these groups (meta and para) were determined using 3D-QSAR models, which showed a positive correlation between these substitutions and an increased inhibitory effect, as evidenced by var573, var383, var164 for ROCK1 and var597 and var477 for ROCK2.
We have added additional explanations in Section 3.1.3. of the manuscript (Design and synthesis of novel ROCK inhibitors), which can be found below:
Lines 708-722: Based on the results of our 3D-QSAR and molecular docking studies for ROCK1 and ROCK2, we designed a series of novel ROCK inhibitors as shown in Figure 6. The hinge binding moieties selected for these compounds include isoquinoline, pyridine, and pyridine-4-phenyl groups, all of which have heterocyclic structures with hydrogen-accepting nitrogen atoms consistent with our computational predictions. These moieties are linked to a scaffold containing both hydrogen-donating (nitrogen of amide) and hydrogen-accepting (carbonyl) groups, consistent with the conclusions identified in our 3D-QSAR models and docking analyses for ROCK1 and ROCK2. Furthermore, the incorporation of a phenyl ring into the linker region increases the potency of the inhibitors by promoting hydrophobic interactions with key residues in ROCK1 (Ala103, Val137, Leu205, Met153, Phe120, Val90 and Gly85) and ROCK2 (P-loop). Our studies have also shown that substituents with hydrogen accepting properties on the phenyl ring further enhance the inhibitory potential of these compounds. To exploit this, we introduced fluoro, hydroxy and methoxy groups at different meta- and para-positions of the phenyl ring. In this way, 9 new potential ROCK inhibitors were developed (Figure 6).
In summary, the structural features of the designed compounds, as shown in Figure 6, are not simple modifications of fasudil, but rather are based on detailed 3D-QSAR and molecular docking studies. These designs reflect key molecular interactions essential for ROCK inhibition and are consistent with the predictive models we developed.
- The enzymatic assays did not compare the activity of C-19 and C-22 with fasudil.
Thank you for your insightful question. In this study, staurosporine was used as standard reference. The ROCK enzyme inhibition assays were performed by Reaction Biology Corporation, who, based on their extensive experience, recommended staurosporine as the standard reference due to its wide range kinase inhibition profile. Given the high similarity between kinases, including the ROCK isoforms, staurosporine serves as an ideal reference compound, providing a more comprehensive assessment of inhibitory potential for a wide range of kinases, including ROCK1 and ROCK2. In contrast, the selectivity of fasudil for ROCK may limit its utility for broader kinase comparison. In addition, staurosporine is an established benchmark for kinase assays that is often used to evaluate the activity, potency and selectivity of novel inhibitors. It allows a more accurate comparison of the inhibitory activity of our compounds.
- The concentration in Figure 12 should be “3.125, 6.25, 12.5”, instead of “3,125, 6,25, 12,5”.
Thank you for pointing out this error. Figure 12 (Figure 10 in the revised manuscript) has now been corrected.
Figure 10. Effect of 72-hour exposure to ROCK inhibitors on the viability of MDA-MB-231, HCC1937, Mia PaCa-2 and Panc-1 cell lines.
- Misspelling in Figure 2 (Ripasudil).
Thank you for your comment. Figure 2 has now been corrected.
- In “3.1.1. Results of 3D-QSAR study for ROCK1 and ROCK2”, authors cited a lot of codes from CHEMBL and describes the presence and absence of certain interactions. It is difficult to understand without structures. Please provide the structures of these compounds in the supporting information.
Thank you for your suggestion. All structures of compounds that are used for 3D-QSAR and molecular docking studies are added now to Supplementary information (Table S3).
- The description of Figure 9 should explain the roles of J0P501 and VFA501.
Thank you for your valuable suggestion. We have revised the description of Figure 9 (Figure 7 in the revised manuscript) to provide a more detailed explanation of the roles of J0P501 and VFA501, as outlined below:
Figure 9. A. The binding position of the novel ROCK inhibitor C-19 (grey) with a ChemPLP score of 80.32 and the co-crystal ligand J0P501 (green) with a ChemPLP score of 82.68 in the active site of ROCK1 (left), together with a 2D representation of key C-19–ROCK1 interactions (right). B. The binding position of C-19 (grey) with a ChemPLP score of 82.82 and the co-crystal ligand VFA501 (blue) with a ChemPLP score of 82.46 in the active site of ROCK2 (left), accompanied by a 2D diagram highlighting the key C-19–ROCK2 interactions. J0P501 and VFA501 are the co-crystal ligands of ROCK1 (PDB: 6E9W) and ROCK2 (PDB: 7JNT), respectively, providing a comparative analysis of their binding modes with those of the new ROCK inhibitor C-19.
Comments on the Quality of English Language
The format of variable should be consistent (some variables are shown in bold, some are not).
Thank you for your comment. All variables are now in bold.

Round 2
Reviewer 2 Report
Comments and Suggestions for Authors
The authors have addressed most of the issues raised in the revision. As a result, the paper is now better in quality.
Reviewer 3 Report
Comments and Suggestions for Authors
This revised version of manuscript has been addressed all issue raised. Therefore, I recommend this article published in this journal.